# Combined influence of oceanic and atmospheric circulations on Greenland Sea Ice concentration

Sourav Chatterjee[1,2]*, Roshin P Raj[3], Laurent Bertino[3], Sebastian H. Mernild[3], Subeesh MP[1], Nuncio Murukesh[1], Muthalagu Ravichandran[1]

[1]National Centre for Polar and Ocean Research, Ministry of Earth Sciences, India

[2] School of Earth, Ocean and Atmospheric Sciences, Goa University, India

[3]Nansen Environmental and Remote Sensing Center and Bjerknes Centre for Climate Research, Bergen, Norway

*Corresponds to*: Sourav Chatterjee (sourav@ncpor.res.in)

**Abstract.**

The amount and spatial extent of Greenland Sea (GS) ice are primarily controlled by the sea ice export across the Fram Strait (FS) and by local seasonal sea ice formation, melting, and sea ice dynamics. In this study, using satellite passive microwave sea ice observations, atmospheric and a coupled ocean-sea ice reanalysis system, TOPAZ4, we show that both the atmospheric and oceanic circulation in the Nordic Seas (NS) act in tandem to explain the SIC variability in the south-western GS. Northerly wind anomalies associated with anomalous low SLP over the NS reduce the sea ice export in the south-western GS due to westward Ekman drift of sea ice. On the other hand, the positive wind stress curl strengthens the cyclonic Greenland Sea Gyre (GSG) circulation in the central GS. An intensified GSG circulation may result in stronger Ekman divergence of surface cold and fresh waters away from the south-western GS. Both of these processes can reduce the freshwater content and weaken the upper ocean stratification in the south-western GS. At the same time, warm and saline Atlantic Water (AW) anomalies are recirculated from the FS region to south-western GS by a stronger GSG circulation. Under a weakly stratified condition, enhanced vertical mixing of these subsurface AW anomalies can warm the surface waters and inhibit new sea ice formation, further reducing the SIC in the south-western GS.

## 1 Introduction

The freshwaters in the GS plays an important part for Nordic Seas overflow (Huang et al., 2020), which constitutes the lower limb of the Atlantic meridional overturning circulation (Chafik and Rossby 2019). The freshwater content in this region is largely driven by the amount of sea ice therein (Aagaard & Carmack 1989). Sea ice in GS is also important in determining shipping routes (Instanes et al. 2005; Johannessen et al. 2007), as well as to the regional marine ecosystem due to its impact on the light availability (Grebmeier et al. 1995). Most of the sea ice in the GS is exported from the central Arctic Ocean across the Fram Strait (FS) and is largely controlled by the ice-drift with the Transpolar Drift (Zamani et al. 2019). Anomalous sea ice export through the FS is associated with events like the 'Great Salinity Anomaly' (Dickson et al. 1988) which can have impact on the freshwater content in the Nordic Seas. Therefore, it is quite evident that the changes in sea ice export through the FS influence the GS sea ice and thus the freshwater availability in the Nordic Seas (Belkin et al. 1998; Dickson et al. 1988; Serreze et al. 2006).

Even though it is one of the main mechanisms contributing to the overall SIC in GS, the relation between sea ice export through FS and SIC variability in GS is not very robust (Kern et al. 2010). This further points to the importance of local sea ice formation and sea ice dynamics in the GS. The impact of these processes can be realized prominently in the marginal ice zone (MIZ) in the south-western GS and the 'Odden' region in central GS (see Fig. 1 for approximate locations of the regions). These regions exhibit strong negative SIC trends during recent decades (Rogers and Hung, 2008, see also Fig. 1a in Selyuzhenok et al. 2020). Changes in sea ice of this region can modify the deep water convection through influencing both the heat and salt budgets (Shuchman et al. 1998). Selyuzhenok et al. (2020) found that in spite of increasing sea ice export through the FS, the overall sea ice volume (SIV) in the GS has been decreasing during the period 1979–2016. They further attributed the interannual variability and decreasing trend of SIV to local oceanic processes, more precisely warmer AW temperatures in the Nordic Seas. Further local meteorological parameters e.g. air temperature, wind speed and direction along with oceanic waves, eddies have also been found to influence the sea ice properties in the central GS, particularly for the Odden region (Campbell et al. 1987; Johannessen et al. 1987; Wadhams et al. 1996; Shuchman et al. 1998; Toudal 1999; Comiso et al., 2001).

Besides the local factors, sea ice in the GS also responds to large-scale atmospheric forcing. For example, a high sea level pressure (SLP) anomaly over the NS results in anomalous southerly wind in the GS. The associated Ekman drift towards the central GS may assist the eastward expansion of the sea ice and SIC increase in the central GS (Germe et al. 2011). Selyuzhenok et al. (2020) also argued that consistent positive North Atlantic Oscillation (NAO) forcing in recent decades have led to warmer AW in the Nordic Seas and resulted in a declining sea ice volume trend. However, the response of Nordic Seas circulation to the atmospheric forcing and the mechanism through which it can influence the SIC in GS is not studied in detail.

The Greenland Sea Gyre (GSG) is a prominent large-scale feature of the Nordic Seas circulation and can be identified as a cyclonic circulation in the central GS basin (Fig. 1). It is known to respond to the atmospheric forcing in the NS and contribute to AW heat distribution in the Nordic Seas (Hatterman et al. 2016; Chatterjee et al. 2018). A stronger GSG circulation increases the AW temperature in the FS by modifying the northward AW transport in its eastern side (Chatterjee et al. 2018). A simultaneous increase in its southward flowing western branch, constituting the southern recirculation pathway of AW (Hattermann et al. 2016; Jeansson et al. 2017), can increase the heat content in the south-western GS through a stronger and warmer recirculation of AW (Chatterjee et al. 2018). The return AW, even after significant modification, remains denser than the local cold and fresh surface waters and thus mostly remain in the subsurface (Schlichtholz & Houssais 1999; Eldevik et al. 2009). However, enhanced vertical winter mixing can cause warming of the surface waters in the GS (Våge et al., 2018). Further, the eastward flowing Jan Mayen Current (JMC), originated from the East Greenland Current (EGC), constitutes the south-western closing branch of the cyclonic GSG circulation in the GS (Fig. 1b). The east-ward extension of the cold and fresh JMC into the central GS basin helps in both new sea ice formation and advection of sea ice from the EGC (Wadhams & Comiso 1999). Changes in GSG circulation and associated AW recirculation in GS may also influence the JMC strength and temperature. Thus given the potential role of GSG in modifying the oceanic conditions, it is important to understand how the response of GSG circulation to the atmospheric forcing can influence the SIC in the GS.

In this study we hypothesize that the interannual winter mean SIC variability in GS can be explained by the combined influence of atmospheric and oceanic circulations, more precisely the GSG circulation. Using a combination of satellite passive microwave SIC, a coupled sea ice ocean reanalysis and atmospheric reanalysis data, we show that changes in the GSG dynamics and resulting AW transport in GS can potentially influence the SIC in the south-western GS. Further, we also show that the atmospheric circulation associated with the GSG circulation variability provides the favourable conditions for the GSG's control on the SIC variability in the south-western GS region. Section 2 describes the data and methods applied in the study following the results in section 3. Discussions and conclusions are mentioned in section 4.

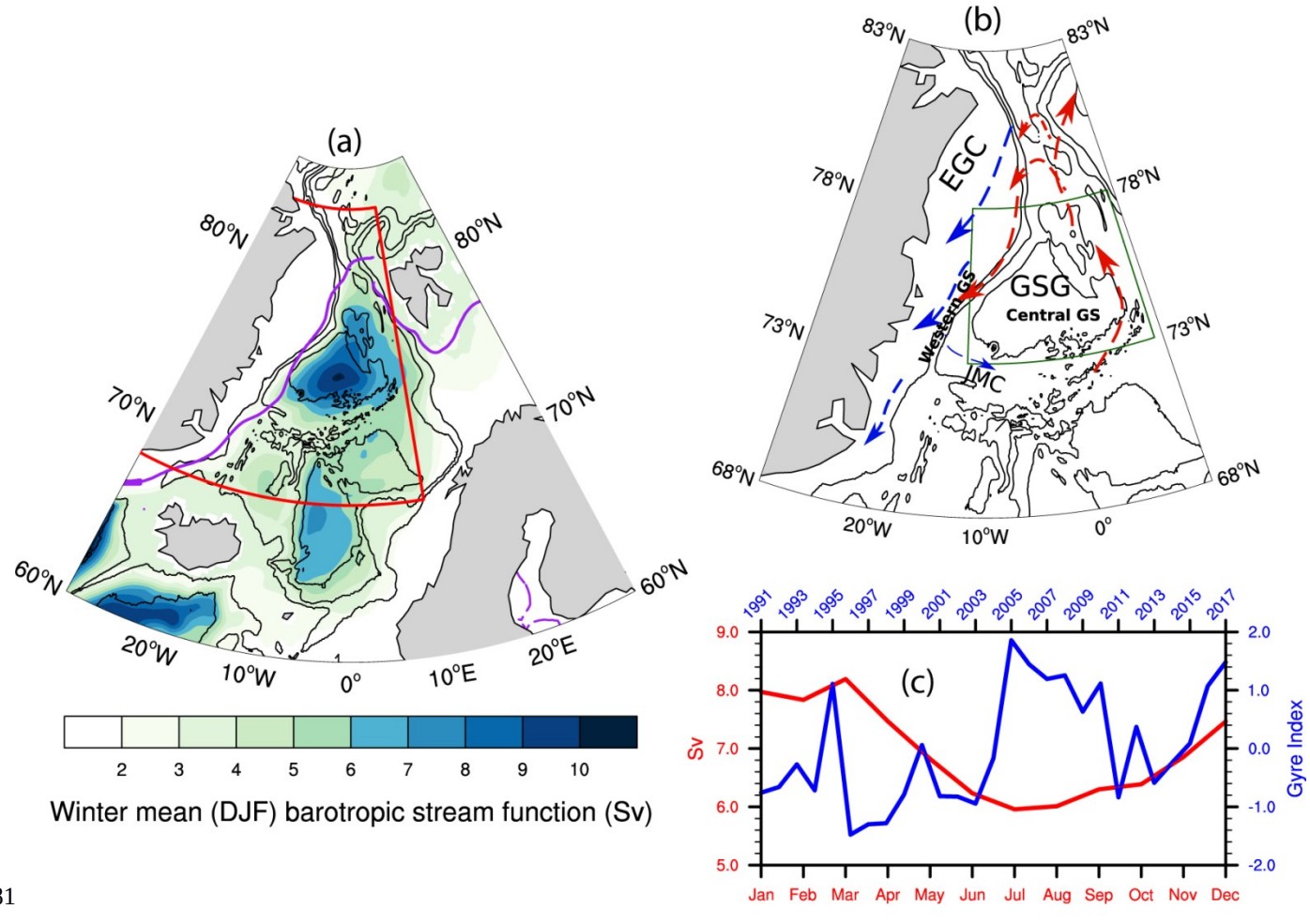

81

82

**Figure 1:** a) Winter mean (DJF) barotropic stream function for the period 1991–2017. The region marked in red indicates the Nordic Seas region. The purple line shows the mean DJF sea ice extent for the study period. b) Schematic of the major currents and discussed in the text. JMC: Jan Mayen Current; EGC: East Greenland Current; GSG: Greenland Sea Gyre. Warm currents are drawn in red and cold currents are in blue. Black contours are showing bottom topography drawn at every 1000 m. The thick black contour indicates the 3000m isobath. The marked region in dark green is used to calculate the 'gyre index' as detailed in the next section. c) The blue line indicates the gyre index used in this study and the red line shows the annual cycle of the strength of GSG circulation determined by averaging barotropic stream function within the 3000m isobath in the region marked in (b).

## 2. Data

### 2.1 Atmospheric data:

Monthly mean sea level pressure (SLP) data was obtained from the ERA Interim reanalysis (Dee et al. 2011) for the period 1991–2017 on a 0.5 by 0.5 degree grid resolution. Monthly anomalies were calculated from the monthly climatology field using the full time period (1991–2017) and were averaged for December-January-February (DJF). For the linear regression analysis the DJF averaged SLP anomalies were detrended.

### 2.2 Oceanic data:

Monthly mean oceanic data used in this study were taken from TOPAZ4, a coupled ocean and sea ice data assimilation system for the North Atlantic and the Arctic. TOPAZ4 is based on the Hybrid Coordinate Ocean Model (HYCOM, with 28 hybrid z-isopycnal layers at a horizontal resolution of 12 to 16 km in the Nordic Seas and the Arctic) and Ensemble Kalman Filter data assimilation, the results of which have been evaluated in earlier studies (Lien et al. 2016; Xie et al. 2017; Chatterjee et al. 2018; Raj et al. 2019). TOPAZ4 represents the Arctic component of the Copernicus Marine Environment Monitoring Service (CMEMS) and is forced by ERA Interim reanalysis and assimilates (every week) observations from different platforms. The detailed setup and performance of the TOPAZ4 reanalysis, including the counts of observations and the temporal variations of the data counts are described in Xie et al. (2017). Of particular relevance for GS are the assimilation of Argo profiles, research cruises CTDs from Institute of Oceanology Polish Academy of Science (IOPAS) and Alfred-Wegener Institute (AWI) (Sakov et al. 2012), satellite sea ice concentration, sea surface temperature and sea level anomaly from the CMEMS platforms.

### 2.3 Sea ice data:

Monthly mean sea ice concentrations (SIC) from Nimbus-7 SMMR and DMSP SSM/I-SSMIS Passive Microwave Data, Version 1 (Cavalieri et al. 1996) were obtained from the National Snow and Ice Data Centre for the period 1991–2017. The dataset provides a continuous time series of SIC on a polar projection at a grid scale size of 25km by 25km. Sea ice velocity data was taken from the Polar Pathfinder Daily 25 km EASE-Grid Sea Ice Motion Vectors (Tschudi et al. 2019).

### 2.4 Methods and Evaluation of TOPAZ4

We estimated the strength of the GSG circulation by area-averaging the winter-mean (DJF) barotropic stream function anomalies within the 3000m isobath in the region 73 N:78 N; 12 W:9 E (as marked with green box in Fig. 1b). The area-averaged values were then standardized over the complete time period 1991–2017 to estimate the 'gyre index' (Fig. 1c). In this study we focused only on the winter (DJF) season as the local sea ice in GS can only form during winter and also the strength of the GSG circulation peaks during winter (Fig. 1c). Composite analysis of DJF mean potential temperature anomaly was performed by averaging the same for strong and weak gyre index years which were determined when the gyre

124index crosses the 0.75 and -0.75 mark respectively. The 0.75 threshold was chosen to consider only the sufficiently
125strong/weak gyre circulation periods. Throughout the article, all regression and correlation analysis were performed with the
126detrended time series for the corresponding variables. Freshwater content was calculated using the following formula

127
$$\int_{z}^{surf} \frac{S_{ref} - S}{S_{ref}} dz$$

128where, $S$ is salinity and the reference salinity $S_{ref}$ is chosen as 34.8 psu.

129

130The standard deviation of winter-mean DJF SIC, in both observation and TOPAZ4, showed high variability along the MIZ in
131south-western GS and the Odden region in central GS (Fig. 2). Note that, the TOPAZ4 reanalysis data exhibits a more
132confined MIZ than observations, which is a known model deficiency (Sakov et al. 2012). The sea ice model (Hunke and
133Dukowicz, 1997), used in TOPAZ4, has a narrower transition zone between the pack ice and the open ocean. Although
134assimilation of the sea ice observations does slightly improve the position of MIZ in TOPAZ4 compared to observation, the
135sharp transition in a narrow band still remains, which could have resulted in higher standard deviations in a narrow MIZ of
136TOPAZ4 as observed in Fig. 2b. However, as we will find in the next section, the sea ice response to the atmospheric and
137oceanic processes explained in the study can be significantly found in both the observation and TOPAZ4 with slightly higher
138signals along the MIZ in TOPAZ4. Thus the higher signal-to-noise ratio in TOPAZ4 should not affect the qualitative aspects
139of the processes and their influence on SIC, which is the main objective of the study.

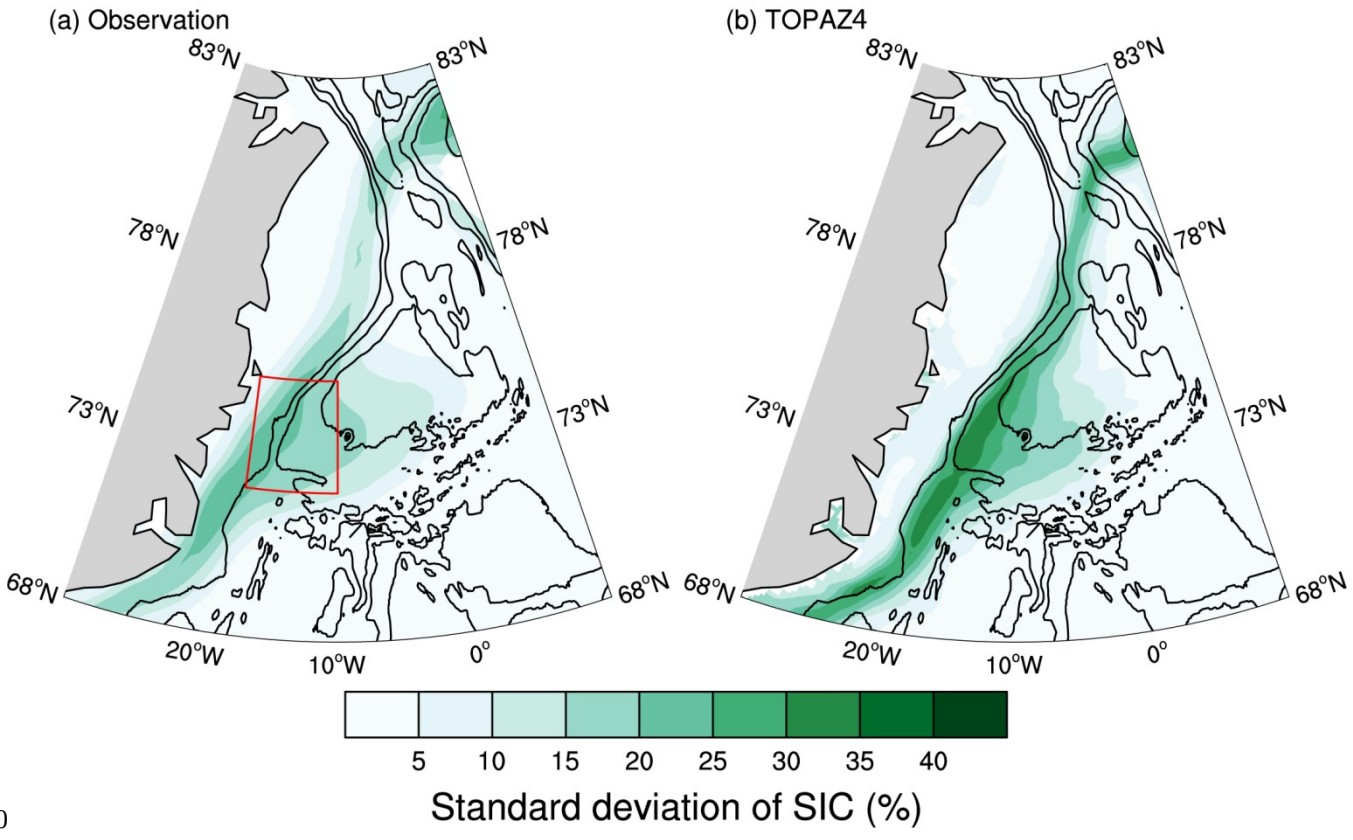

142**Figure 2:** Standard deviations of DJF monthly mean sea ice concentration for the period 1991–2017 from (a) satellite 143observations (b) TOPAZ4 reanalysis. The red box with high values is drawn over the region 72N:75N; 18W:10W and is 144referred to as south-western GS hereafter.

145For evaluation of the oceanic conditions in TOPAZ4 we used temperature and salinity observations obtained from EN4 146(version 4.2.1) quality controlled analyses with Levitus et al. (2009) corrections applied. Here we chose to compare the 147oceanic parameters in a region (as marked in Fig. 2) in south-western GS where the standard deviation of the SIC is found to 148be maximum both in TOPAZ4 and observations. Also we will show in the next section that SIC response to the processes 149described here is most profound in this region. Hereafter we refer to this region as south-western GS for simplicity. Fig. 3 150shows the spatio-temporal patterns of sea surface temperature (SST) and salinity (SSS) in south-western GS as found in 151TOPAZ4 and EN4. Although the temporal evolution of these parameters are well captured in TOPAZ4, compared to 152observation, the westward extension of the warm and saline waters was found to be less in TOPAZ4. This indicates that the 153front between the cold and fresh waters along the Greenland shelf and the warm and saline waters in the south-western GS is 154slightly shifted towards the east in TOPAZ4 compared to observation. This could be a reason for the fact that higher standard

155deviation of SIC is found slightly toward the east in TOPAZ4 than observations (Fig. 2). In south-western GS, both the
156surface and subsurface temperature in TOPAZ4 was found to be colder compared to observations (Fig. 4). The negative
157biases in TOPAZ4 were more profound in the subsurface for both temperature and salinity. Xie et al., (2017) also found a
158similar result with TOPAZ4 and attributed it to sparse observations. Using the potential density difference between 200m
159and the surface as an indicator of the stratification, we found that TOPAZ4 has weaker stratification compared to
160observations (Fig. 4e). Consistent with the cold bias in TOPAZ4, winter-mean SIC in TOPAZ4 is higher than the satellite
161observation in the south-western GS (Fig. 4f). However, we found a strong correlation (r=0.9) between the SIC in
162observation and TOPAZ4. This indicates that the interannual variability of SIC, which is the focus of the study, is quite
163consistent in both TOPAZ4 and observation.

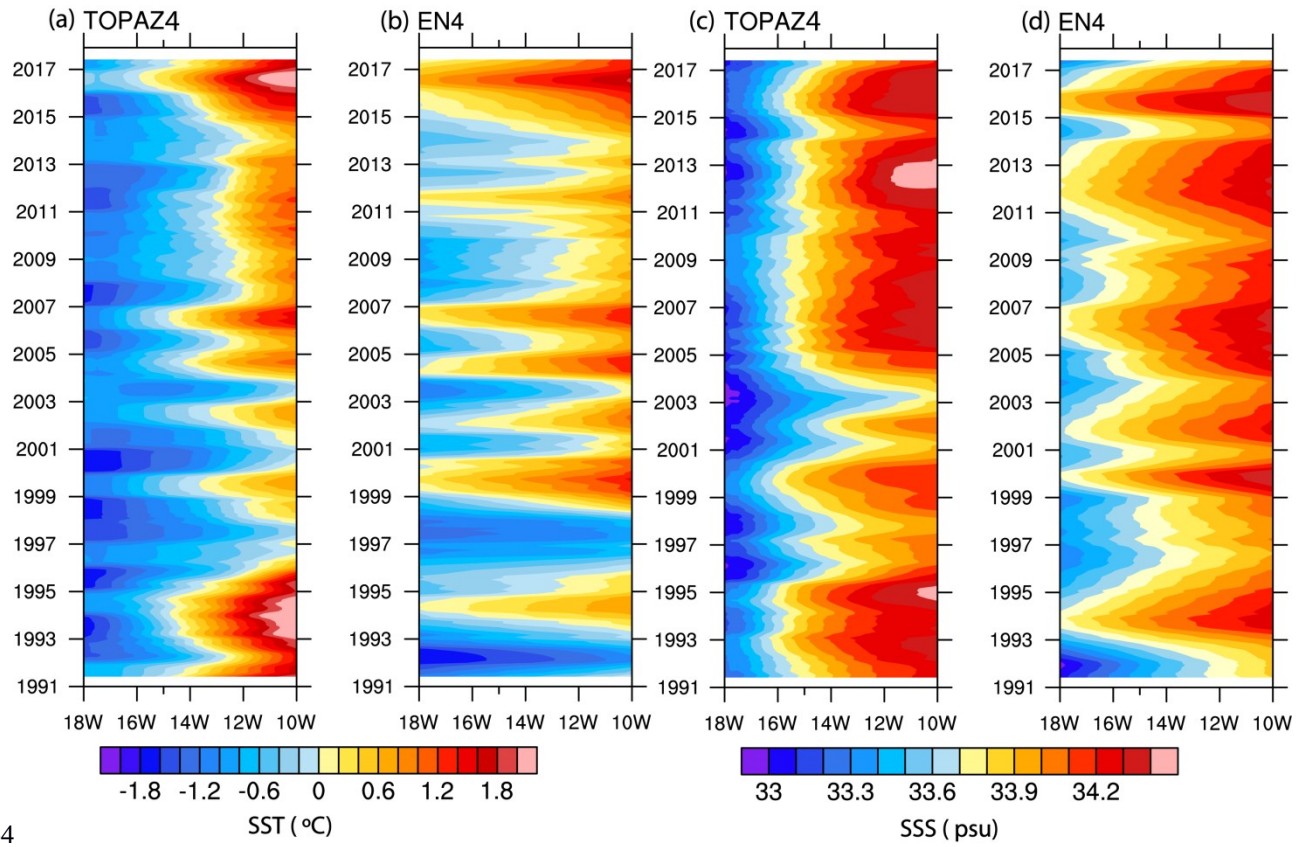

164

165**Figure 3:** Hovmoller (longitude-time) diagram of the SST (°C; a,b) and SSS (psu; c,d) over the region over 72 N:75 N; 18
166W:10 W in the south-western GS as marked in Fig. 2. (a) and (c) are for TOPAZ4 and (b) and (d) for EN4 observations. In
167all cases data were smoothed with one year running mean.

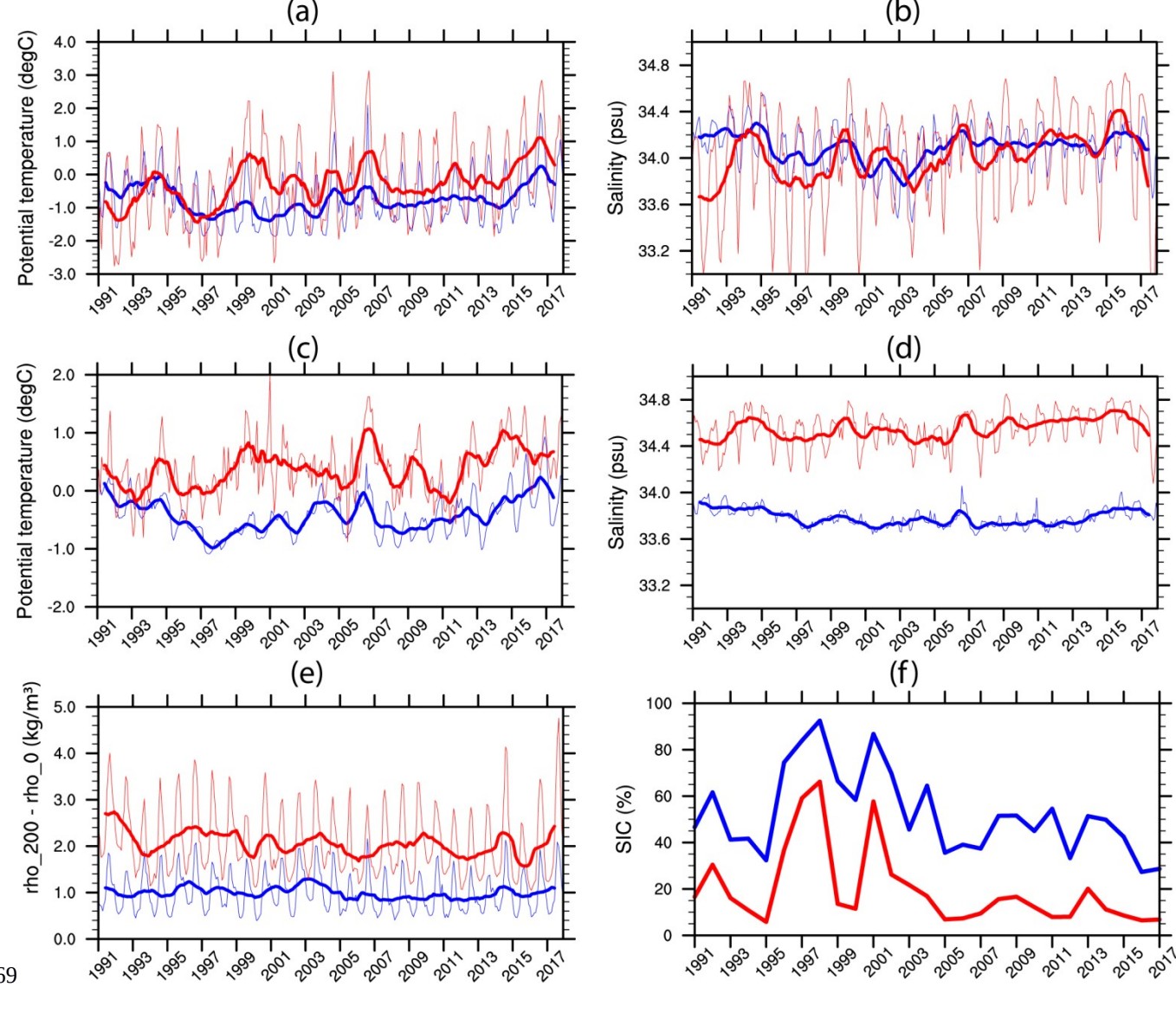

170

**Figure 4**: Comparison between EN4 observation (red lines) and TOPAZ4 (blue lines). Monthly mean (thin lines) and one year running mean (thick lines) of potential temperature (a,c), salinity (b,d) and stratification index (e, difference of potential density between 200m and surface) averaged over 72 N:75 N; 18 W:10 W in the south-western GS as marked in Fig. 2. (a,b) are for 0-50m depth average and (c,d) for 100-400m depth average. (f) DJF mean sea ice concentration in the same region from satellite observation (red) and TOPAZ4 (blue).

## 3. Results

The regression map of winter mean SIC on the gyre index showed significant negative SIC in the south-western GS (Fig. 5). The spatial pattern of the regression coefficients closely resembles the standard deviation of winter mean SIC in the GS, as shown in Fig. 2. This indicates that a considerable amount of the SIC variability in GS can be associated with GSG circulation. However, it should be noted that the atmospheric forcing in the NS can influence both the GSG circulation (Aagaard 1970; Legutke 2002; Chatterjee et al. 2018) and SIC variability in the GS (Germe et al. 2011).

183

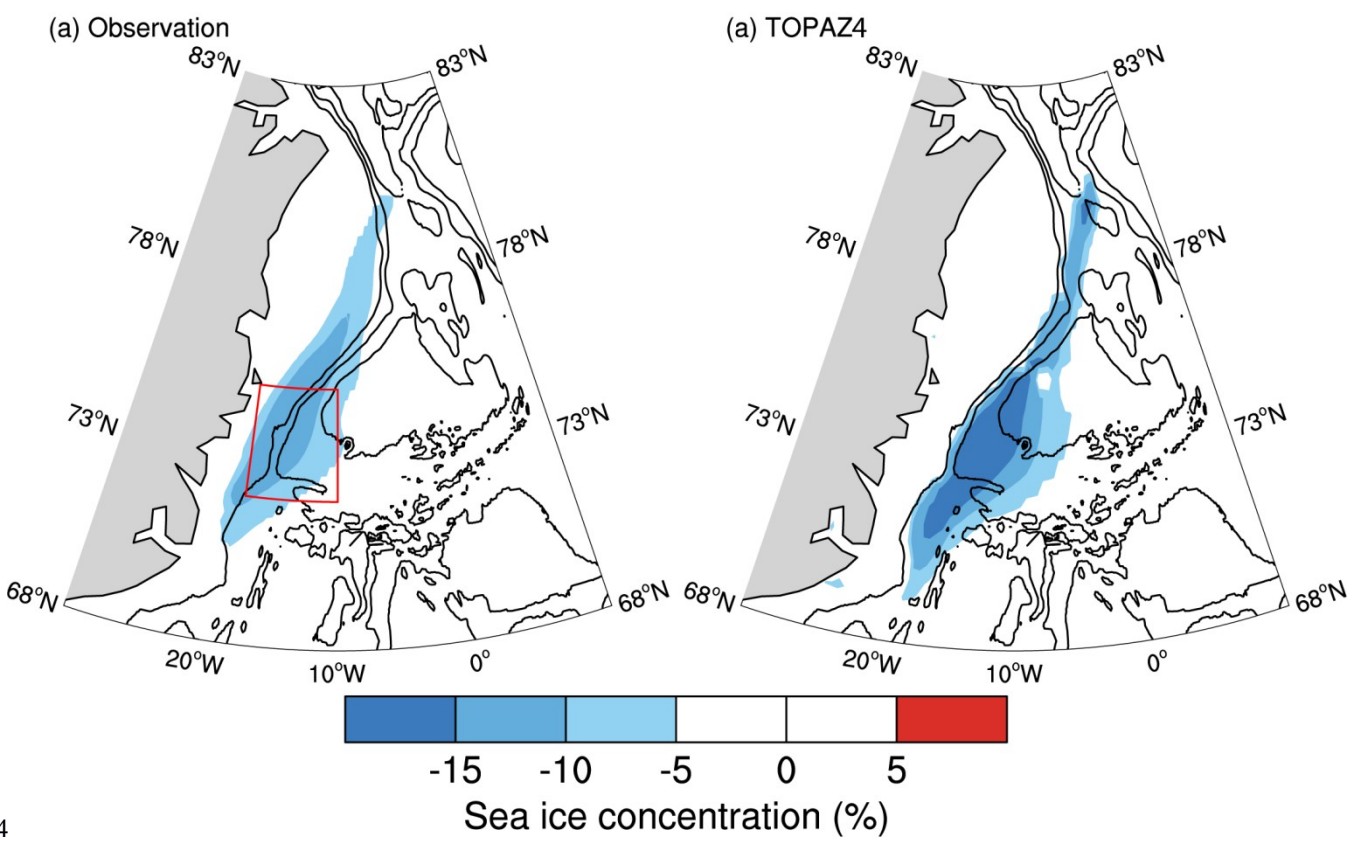

**Figure 5:** Linear regression of winter mean (DJF) sea ice concentration from (a) satellite observation (b) TOPAZ reanalysis on the gyre index. Only significant values at 95 % level are shown. Contours are bottom topography drawn at every 1000 m.

189To elucidate the possible influence of atmospheric circulation pattern associated with GSG circulation on the SIC variability 190in the GS, linear regression of the sea level pressure anomalies on the gyre index was calculated and shown in Fig. 6. The 191large-scale atmospheric circulation shows a positive NAO-like pattern associated with a strong GSG circulation, but with 192centres of actions north of their usual locations (Fig. 6). The GSG circulation responds to the anomalous wind stress curl 193induced by the low SLP anomaly patterns in the NS (Chatterjee et al. 2018). However, we found that the station based NAO 194index, with its spatial feature highlighting the Icelandic low and Azores high, (https://climatedataguide.ucar.edu/sites/default/ 195files/nao_station_seasonal.txt) and the gyre index have a very low correlation (r = 0.2). This further points to the importance 196of the spatial variability of NAO (Zhang et al. 2008; Moore et al. 2012) and its influence on the Nordic Seas circulation. 197Also note that the low correlation could be due to the fact that the equatorward pole of NAO doesn't exhibit much significant 198regression patterns in Fig. 6.

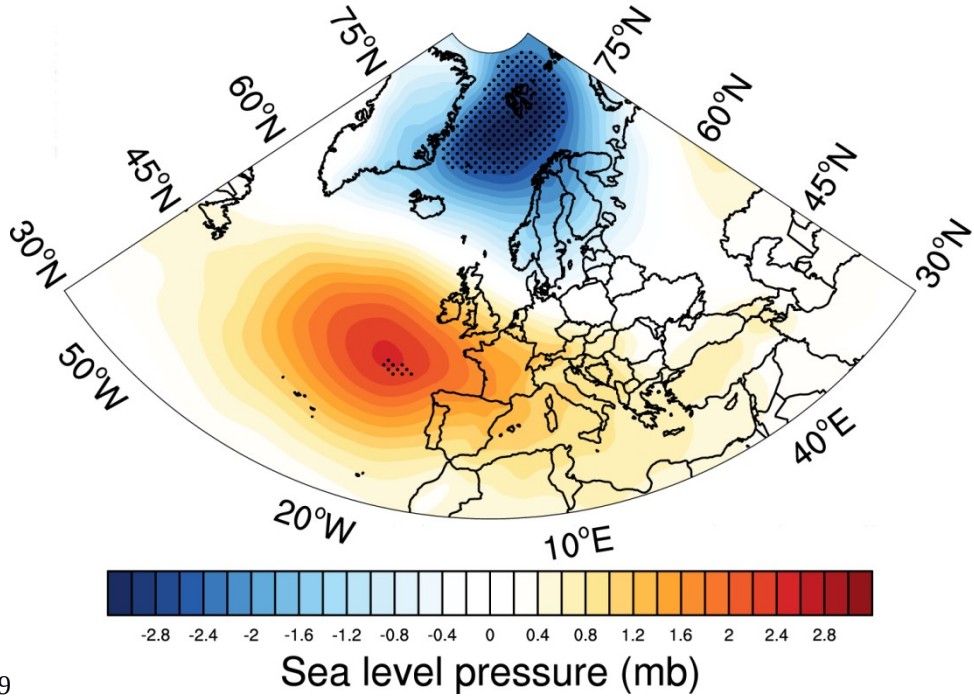

199

200**Figure 6:** Linear regression of DJF mean sea level pressure anomaly on the gyre index. Regions with 95% statistical 201significance are dotted.

202The mean southward sea ice export in the GS across the FS (Fig 7a) is strongly driven by the geostrophic winds in this 203region (Smedsrud et al. 2017). The low SLP pattern over NS associated with the GSG circulation can induce anomalous 204northerlies in GS. Linear regression of sea ice velocities on the gyre index showed anomalous northward sea ice velocities in 205GS associated with increase in GSG strength (Fig. 7b). This indicates that the anomalous northerly winds during a strong 206GSG circulation would lead to Ekman drift of sea ice which tends to push the sea ice towards the Greenland coast and reduce

12                                    12

207the mean southward sea ice velocities in this region (Fig. 7a). This could lead to reduced sea ice export in this region and
208result in low SIC.

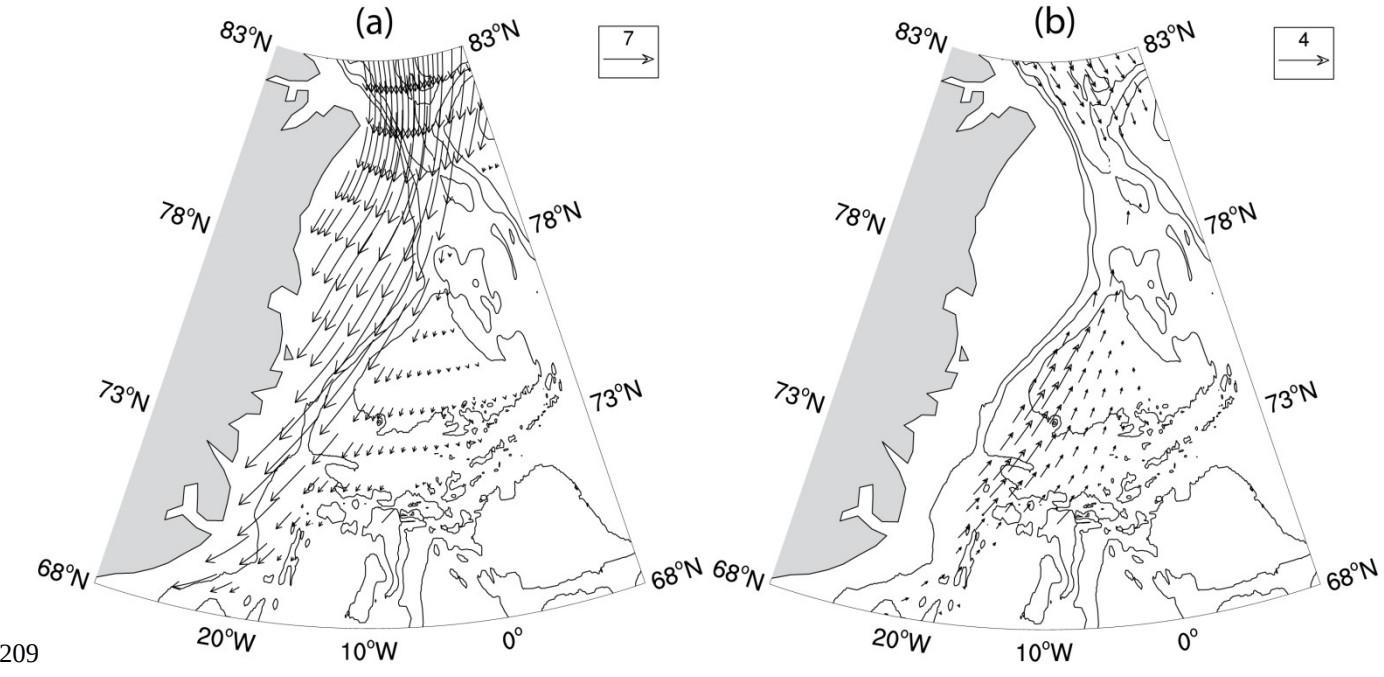

209

210

211**Figure 7:** (a) Climatological (1991–2017) DJF sea ice velocity vectors (cm/s) from satellite observations. (b) Regression of
212DJF sea ice velocity anomalies (cm/s) on the gyre index. Only results significant at 95 % are shown for clarity. Contours are
213bottom topography drawn at every 1000 m.

214Next, we investigate GSG's potential in influencing the oceanic conditions and hence the sea ice in the GS, given that the
215local oceanic conditions largely affect the sea ice conditions therein (Johannessen et al. 1987; Visbeck et al. 1995; Kern et al.
2162010; Selyuzhenok et al. 2020). Figure 8a shows the difference in ocean temperature anomaly in the upper 400m averaged
217for the strong and weak GSG circulation years (marked in Fig 8b; see methods for definitions). The average temperature
218anomaly for the strong GSG circulation years was found to be ~1°C higher than the same during weak GSG circulation
219years. The warm anomalies further extend eastward with the JMC towards the central GS and could potentially affect the sea
220ice formation in the Odden region. Further, we found significant positive correlation (r=0.7, p<0.01; Fig 8b) between gyre
221index and temperature advection ($U.\nabla T$ in upper 400m) in the south-western GS (marked region in Fig. 8a), where
222maximum GSG influence on SIC is found (Fig. 3a). This suggests that a strong GSG circulation recirculates the warm AW
223anomalies into the south-western GS from the FS. This is consistent with earlier study indicating an increased oceanic heat
224content in the south-western GS due to a stronger GSG circulation (Chatterjee et al., 2018).

13                                          13

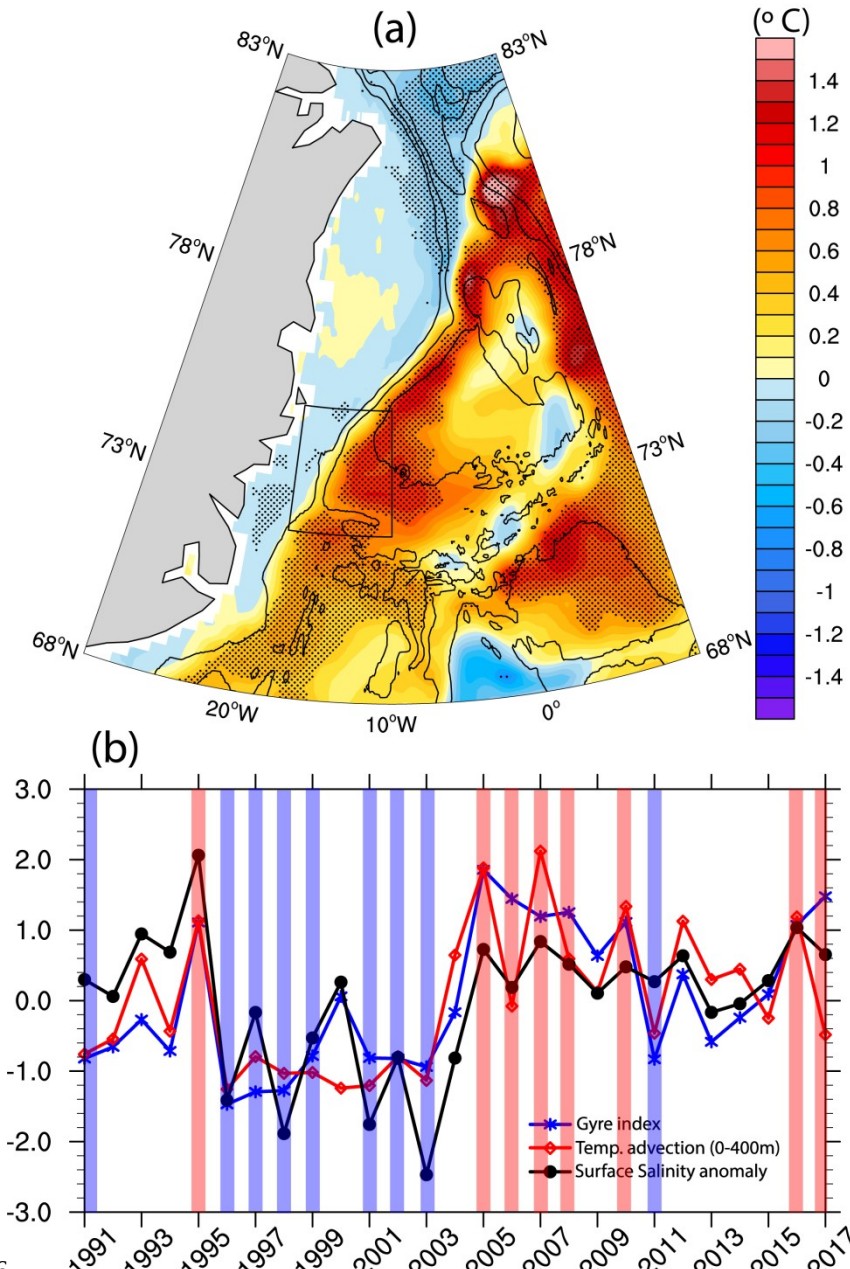

**Figure 8:** (a) Difference between 400 m depth averaged potential temperature anomalies (ºC) averaged for strong (red bars in (b)) and weak (blue bars in (b)) gyre index years. (b) Gyre index (blue), and standardized surface salinity anomaly (black), temperature advection ($U . \nabla T$) in upper 400 m (red) for DJF over the region 72 N: 75 N; 18 W : 10 W, as marked in (a).

230However, it should be noted that the recirculated AW in the GS still remains dense enough to be in subsurface ( Schlichtholz
231& Houssais 1999; Eldevik et al.  2009) and needs to be vertically mixed to have an impact on the sea ice. We found that the
232upper ocean stratification in the south-western GS strongly covaries with GSG circulation strength (Fig. 9a). The analysis
233shows that a weakening of the stratification in the upper part of the water column coincides with a stronger GSG circulation
234and vice versa (Fig. 9a). Further, warm and saline signatures in the upper ocean can be found during strong GSG circulation,
235indicating enhanced vertical mixing of the AW in the south-western GS (Figs. 9b,c). This is further confirmed by significant
236positive correlation (r=0.7, p<0.01) between surface salinity anomaly and gyre index (Fig. 8b). These surface anomalies can
237further inhibit new sea ice formation and also may cause melting of existing sea ice from the bottom.

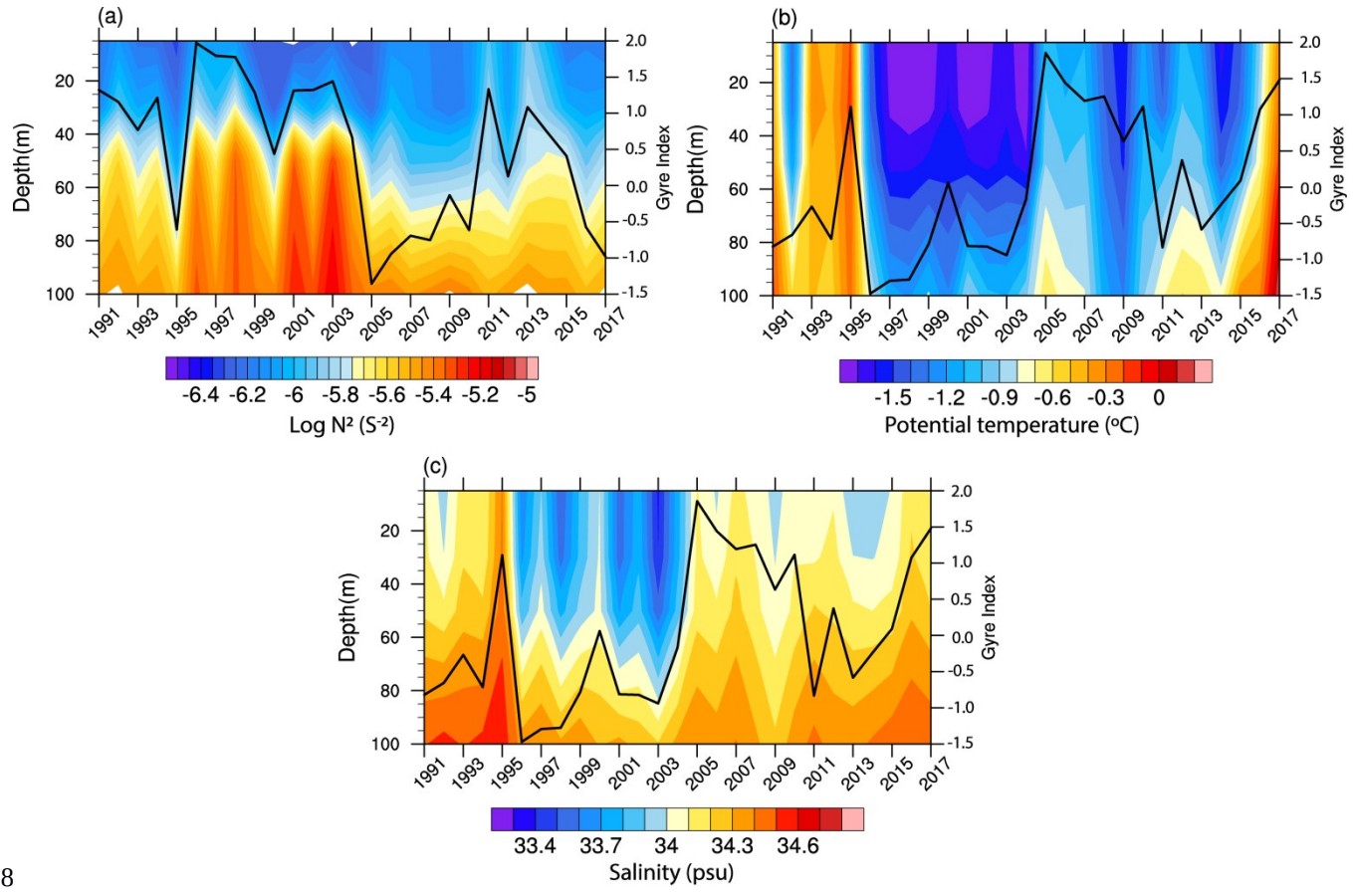

238

239

240**Figure 9:** (a) Logarithm of squared Brunt-Väisälä Frequency (N², colour shaded) (b) potential temperature (c) salinity for
241DJF over the region 72 N:75 N; 18 W:10 W, as marked in Figure 8a. The black timeseries against the right Y axis is the gyre
242index in all three panels. Note that the gyre index is plotted against a reversed Y axis in (a) for ease of comparison.

15                                                                15

**244 4. Discussions and Conclusions**

245 Here we investigated the combined influence of atmospheric and oceanic circulations on the interannual variability of the 246 winter mean SIC variability in the GS and showed that NS, in particular the GSG circulation can significantly contribute to 247 the SIC variability in south-western GS. Fig. 10 shows the flow chart and a schematic illustration of the mechanisms 248 proposed in this study. The large-scale atmospheric circulation pattern that influences the GSG circulation resembles a 249 NAO-like pattern with its northern centre of action situated northeast of the typical NAO pattern. The cyclonic GSG 250 circulation strengthens in response to the positive wind stress curl induced by the low SLP anomaly in the NS (Legutke 251 2002, Chatterjee et al. 2018). The resulting northerly wind anomalies over GS can potentially alter the sea ice export across 252 the FS (Kwok & Rothrock 1999; Jung & Hilmer 2001; Vinje 2001; Tsukernik et al. 2010; Smedsrud et al. 2011; Ionita et al. 253 2016). However, winter mean SIC in the GS and FS ice area flux are not strongly correlated (Kwok et al., 2004; Germe et 254 al., 2011), suggesting that the SIC variability in the GS can be significantly influenced by the local sea ice dynamics and 255 oceanic conditions.

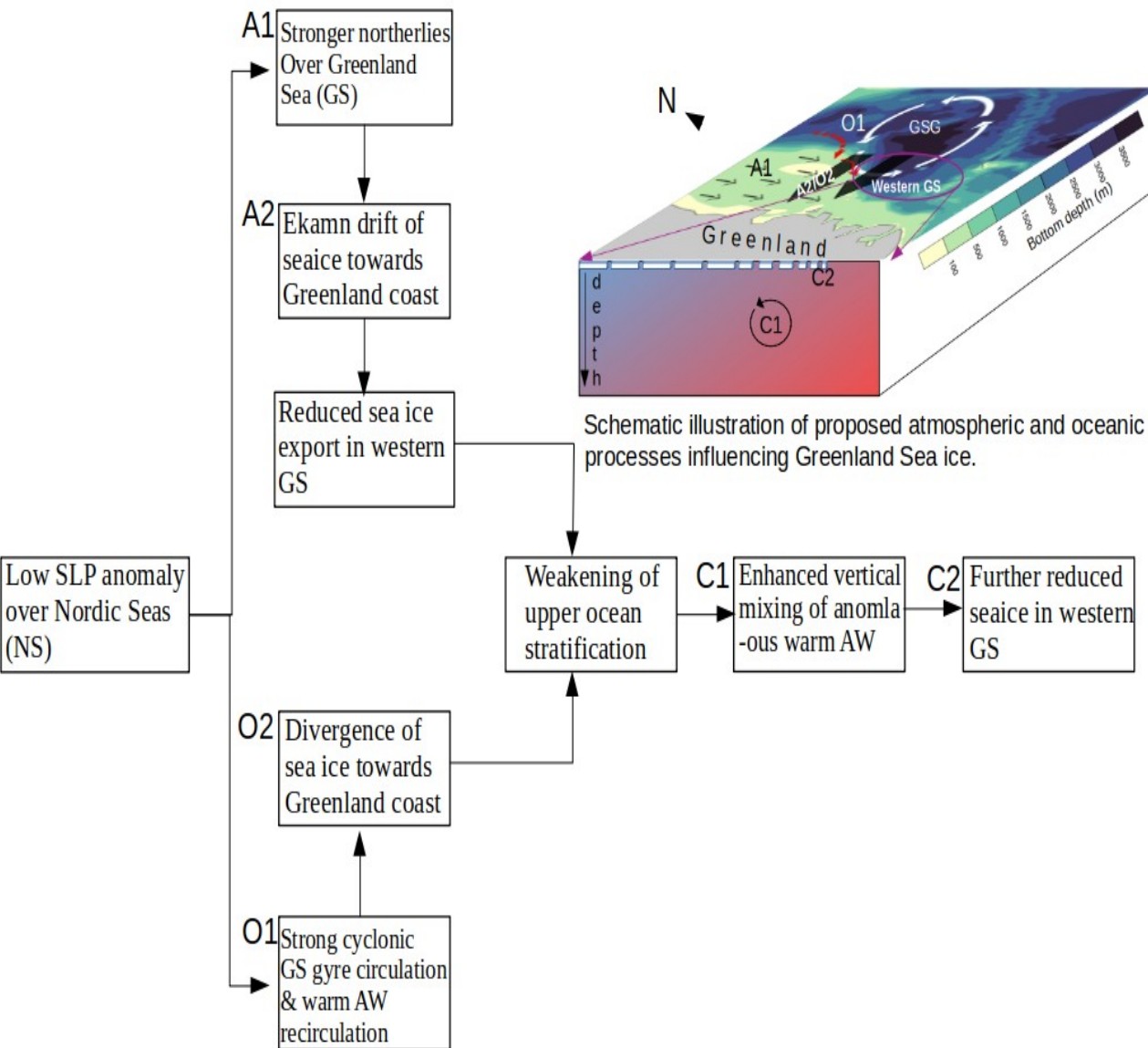

Schematic illustration of proposed atmospheric and oceanic processes influencing Greenland Sea ice.

256

257

258

259**Figure 10:** A flow chart and schematic diagram of the proposed processes influencing the SIC variability in the south-260western GS.

261

262Anomalous winds in the Nordic Seas are known to influence the SIC in the GS through Ekman drift of the sea ice (Germe et
263al., 2011). During time-periods with anomalously low SLP over NS, anomalous northerly winds and associated Ekman drift
264towards the Greenland coast that can reduce the sea ice export in the western and central GS (Fig 8b). Enhanced Ekman
265divergence due to a strengthened GSG circulation can further lead to reduced freshwater and sea ice in the south-western GS
266(Fig. 11). We found that these can lead to weakening of the upper ocean stratification in the south-western GS (Fig. 9a). At
267the same time, a stronger GSG circulation recirculates the warm and saline subsurface AW anomalies from the FS into the
268south-western GS (Fig 8a). These AW anomalies can warm the surface waters by enhanced vertical mixing in a weakly
269stratified condition (Fig. 9) and can cause further reduction of SIC by inhibiting new sea ice formation or even melting the
270sea ice from bottom. Although our study doesn't show bottom melting of the sea ice, this can be realized from the findings
271by Ivanova et al. (2011) which showed enhanced bottom melting in this region during positive NAO periods. Thus, the SIC
272variability in the south-western GS responds to simultaneous influences from the atmospheric and oceanic circulation (Fig.
27310). Despite the known influences of smaller scale processes, such as eddies and wave interactions on the SIC in the south-
274western GS, our results show that the larger scale processes can also significantly affect the SIC variability in the region,
275particularly on interannual timescales when the impacts of smaller scale processes can cancel out or may not be strong
276enough to dampen the impact of larger scale processes. However, as found in Raj et al., (2020) interactions between the gyre
277circulation and the eddies can be an important factor controlling the oceanic conditions and hence the SIC in the south-
278western GS.

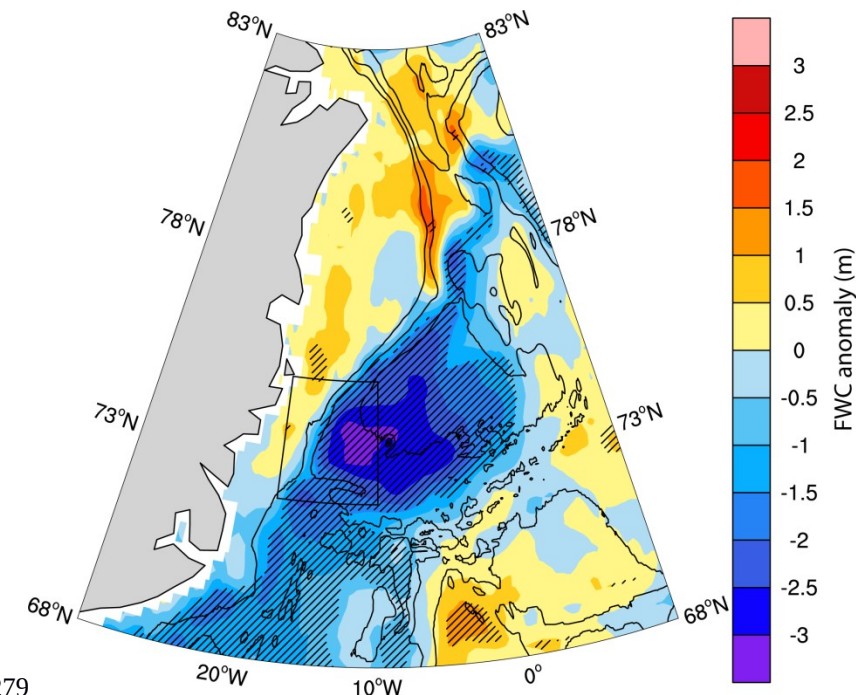

279

280**Figure 11:** Difference in freshwater content (FWC) anomaly (m) between strong and weak gyre index periods. Significant 281differences at 95% level are stippled.

282

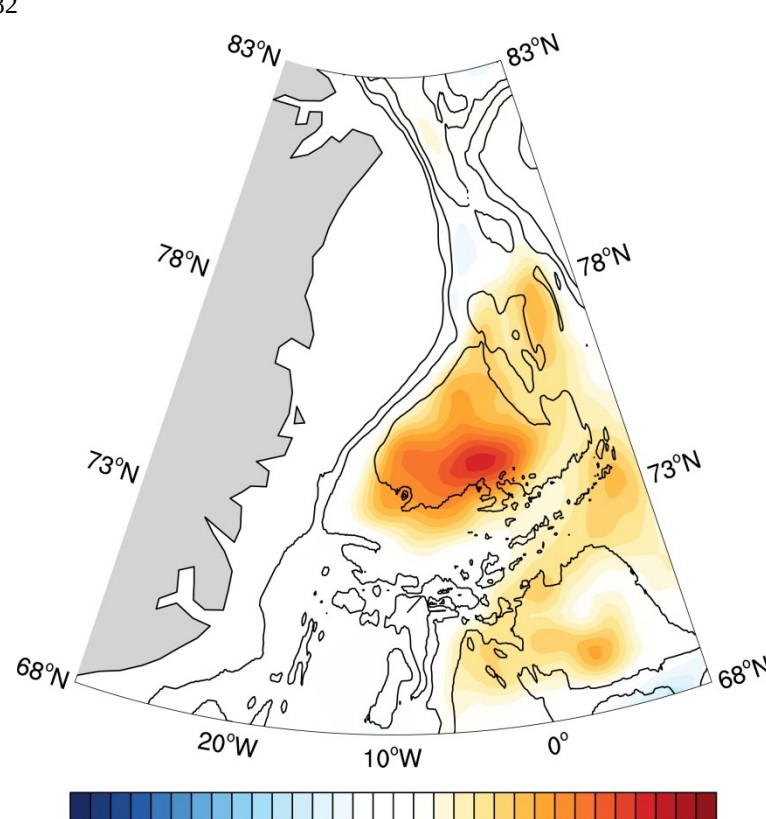

283 Trend in DJF mean Barotropic streamfunction (Sv/yr)

284

285**Figure 12:** Linear trend (Sv/year) in winter mean (DJF) barotropic stream function for 1991–2017. Only significant values at 28695 % level are shown for clarity. Contours are bottom topography drawn at every 1000 m.

287

288 This study finds one of the mechanisms of SIC varibility in the GS, highlighting the role of large scale atmospheric and 289oceanic circulations in the NS. Observations and modelling results suggest stronger atmospheric forcing in the NS due to 290spatial variation of the NAO (Zhang et al. 2008) and its tendency towards positive phase in a warmer climate (Bader et al. 2912011; Stephenson et al. 2016). Consistent with that we find a significant positive trend in the GSG circulation strength 292during the study period (Fig. 12). The response of GSG circulation to this altered atmospheric forcing can further be realized 293with increased GSG strength (Fig. 1c) and a northeastward displacement of NAO's poleward centre of action in the Nordic 294Seas during early 2000s (Fig. 1a in Zhang et al., 2008). Recent observations further suggest intensified convection in the

295GSG and changes in water mass formation during the last two decades (Lauvset et al., 2018; Brakstad et al., 2019). Lauvset 296et al., (2018) further discussed the role of recirculated AW on inducing intensified convection in the GSG through surface 297salinity anomaly. Consistent with this, our results show that the salinity anomalies and intensified convection in the GSG can 298be induced by a stronger GSG circulation (in response to the atmospheric forcing) which helps in recirculation of AW 299anomalies in the GS. Thus we propose that the atmospheric forcing over the NS imposes a positive oceanic feedback (Fig. 30013). The low SLP anomaly over the NS strengthens the GSG circulation. The Ekman divergence pushes the freshwater and 301sea ice from the GS interior towards the coast. Enhanced AW recirculation due to a stronger GSG and weakened 302stratification due to reduced freshwater allows the warm and saline AW anomalies to get vertically mixed and increase the 303temperature and salinity in the central GS. The increased salinity further helps in a stronger GSG circulation, completing the 304feedback loop. However it should be noted that the complex subsrface processes and their interactions with large scale 305circulation are often difficult to capture in the reanalysis, particularly with sparse and interrupted subsurface observations 306over time and space. For example, while the surface variables are well captured in TOPAZ4, it has some limitations with the 307subsurface properties as observed in Xie et. al, 2017. Of particular interest in this study, the south-south-western GS, is an 308exceptionally observational data sparse region. Increased long-term observations from these areas will be helpful in 309improvement of the reanalysis datasets and better understanding of the complex atmosphere-ocean interaction processes and 310their impact on the sea ice variability of this region.

311

312

313

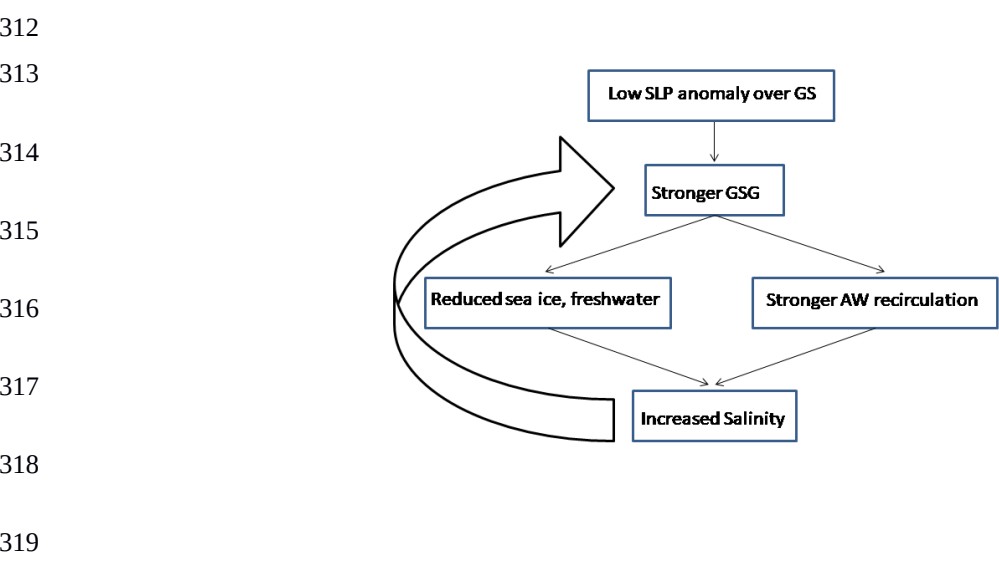

314

315

316

317

318

319

320

321**Figure 13:** A proposed positive oceanic feedback induced by atmospheric forcing in NS.

322

## Acknowledgments

Sea ice concentration (https://nsidc.org/data/NSIDC-0051/versions/1) and velocity (https://nsidc.org/data/nsidc-0116/versions/4) are obtained from the National Snow and Ice Data Centre. The TOPAZ4 simulations have used grants of computing time (nn2993k) and storage (ns2993k) from the Sigma2 infrastructure. The monthly TOPAZ4 results used in this study are obtained via CMEMS (marine.copernicus.eu). EN4 (version 4.2.1) observational data is provided by UK Met Office Hadley Centre and obtained from https://www.metoffice.gov.uk/hadobs/en4/download-en4-2-1.html. Authors thank Ola M Johannessen, Nansen Scientific Society for valuable suggestions during the course of the study. NCPOR is an anutonomous institute fully funded by Ministry of Earth Sciences, Govt. of India. This is NCPOR contribution number J-97/2020-21. All figures were made using The NCAR Command Language (Version 6.4.0).

## Declarations

**Funding** (information that explains whether and by whom the research was supported)

Not Applicable

**Conflicts of interest/Competing interests** (include appropriate disclosures)

Authors declare no Conflicts of interest/Competing interests

**Availability of data and material** (data transparency)

All the data used here are freely available on respective data portals (links provided in the 'Acknowledgements' section)

**Code availability (software application or custom code)**

All the codes are available on reasonable request to the corresponding author.

**Authors' contributions**

SC conceived the idea in discussion with RPR and wrote the manuscript. SC performed all the analyses. All authors contributed in improvement and writing of the manuscript.

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
