# Peer review of "Combined influence of oceanic and atmospheric circulations on Greenland Sea Ice concentration"

_The Cryosphere, 2020_

## Referee Comment (RC1) · Anonymous Referee #1 · 22 Aug 2020

tc-2020-127

**The impact of atmospheric and oceanic circulations on the Greenland Sea ice concentration**
by Sourav Chatterjee, Roshin P. Raj, Laurent Bertino, Sebastian H. Mernild, Nuncio Murukesh, and Muthalagu Ravichandran

In this manuscript the interannual variability of sea ice concentration in the Greenland Sea is investigated. The authors identify several atmospheric and oceanic processes that influence the sea ice concentration, and clarify how these are modulated by the large-scale atmospheric conditions. The authors conclude that the magnitude of the Greenland Sea Gyre circulation is of particular importance.

I think this is an interesting manuscript that highlights the importance of changing sea ice concentrations in the Greenland Sea. My main concern is that the changes in sea ice and ocean conditions over the period considered (1991-2017) are more appropriately characterized by secular trends than interannual variability, in particular reduced sea ice concentration and a warming ocean. I think the variability investigated in this manuscript needs to be discussed within the context of these long-term trends. As such, I recommend that the paper be revised before publication.

**Major comment:**

Sea ice concentration in the western Nordic Seas has steadily diminished over the past decades (Moore *et al.*, 2015; Onarheim *et al.*, 2018). In particular, the Odden ice tongue has rarely formed since the 1990s (e.g. Rogers and Hung, 2008). Over the same period equally remarkable changes in stratification and water mass transformation in the Greenland Sea have taken place (Ronski and Budéus, 2005; Latarius and Quadfasel, 2016; Lauvset *et al.*, 2018; Brakstad *et al.*, 2019). Yet these secular trends, which dominate the variability investigated in the manuscript, are barely, if at all, mentioned. This is important context that needs to be discussed and accounted for.

**Specific comments:**

Line 29:
The first two sentences of the introduction are too strong. As the source of dense overflow waters that supply the deep limb of the Atlantic Meridional Overturning Circulation, the Nordic Seas are indeed very important (e.g. Chafik and Rossby, 2019). But for the Greenland Sea to control regional and hemispheric climate, the Greenland Sea would have to be a main source of overflow water. This is likely not the case (Mauritzen, 1996; Eldevik *et al.*, 2009).

Lines 37-45:
Please clarify that the Odden ice feature rarely developed after the 1990s (e.g. Rogers and Hung, 2008) and that in the present climate only intermediate (not deep) waters are formed in the Greenland Sea (e.g. Brakstad *et al.*, 2019).

Line 57:
Please clarify where the high sea level pressure anomaly patter would have to be located in order to result in anomalous southerly wind in the Greenland Sea.

Line 70:

Please clarify how the Greenland Sea Gyre contributes to heat distribution in the Nordic Seas. It seems more plausible that heat inside a gyre would be trapped rather than distributed.

Line 74:

Please also clarify to what extent a strengthened western branch of the Greenland Sea Gyre circulation results in increasing Atlantic Water transport into the central Greenland Sea vs. Atlantic Water throughput as part of the East Greenland Current (Woodgate *et al.*, 1999). The bulk of the Atlantic Water remains within the East Greenland Current and is transported toward Denmark Strait (Håvik *et al.*, 2017).

Line 108:

Has TOPAZ been evaluated against observations in the central Greenland Sea? Latarius and Quadfasel (2016) or Brakstad *et al.* (2019) would be good points of comparison.

Line 142:

Does the regression map show significant negative sea ice concentration in the central Greenland Sea when the Greenland Sea Gyre is strong?

Line 155:

Please clarify that it (presumably) is the large-scale atmospheric circulation associated with the Greenland Sea Gyre circulation that features an NAO-like pattern.

Line 159:

It is unclear how the low correlation between the gyre and NAO indices signifies an importance of NAO on the circulation in the Greenland Sea.

Line 161:

Please clarify how winds influence the drift of sea ice. Is the drift primarily determined by Ekman transport or directly by the wind? To what extent does that depend on sea ice concentration?

Line 172:

The statement that wintertime Greenland Sea sea ice concentration and Fram Strait ice are flux are not strongly correlated appears to directly contradict the statement on line 35 that changes in ice export through Fram Strait influence the Greenland Sea sea ice concentration.

Line 188:

Does the gyre bring Atlantic Water into the central Greenland Sea or circulate Atlantic Water around the periphery of the Greenland Sea?

Line 230:

Please expand on how Atlantic Water anomalies would impact sea ice formation and how that may influence convection.

Line 251:

The central Greenland Sea has largely been ice free since the 1990s (Moore *et al.*, 2015; Brakstad *et al.*, 2019). This large-scale sea ice retreat, consistent with sea ice loss across the entire Arctic region, is likely not related to the magnitude of the Greenland Sea Gyre circulation.

Figure 1:

The black oval indicating the central Greenland Sea extends onto the Greenland shelf, which should not be considered part of the central Greenland Sea.

Figure 2:

Is it reasonable to average sea ice concentration over the entire 1991-2017 period if the variability is dominated by an ice Odden "on" or "off" state? To the extent that the Odden feature is binary (on or off), the average would represent an in-between state that is never realized. Perhaps consider comparing instead observations and TOPAZ for years when Odden is present and for years when it is not.

Figure 6:

What are the correlations between salinity anomaly, temperature advection, and gyre index? Please clarify in the caption what temperature advection means. Is it heat transport or a product of temperature and velocity, and where is it evaluated?

Figure 7:

It appears that one buoyancy frequency profile per year is shown in Fig. 7. Are the values annual means or summertime means? Please clarify. For most of the year the mixed-layer depth in the Greenland Sea is deeper than 50 m and the buoyancy frequency would be very low. I think it would be sensible to consider the stratification to a deeper level in the Greenland Sea. These days buoyancy frequency seems to be more commonly used than Brunt-Väisälä frequency.

**Detailed comments:**

Line 32:

It should be "... from the central Arctic **Ocean** ..."

Line 59:

Although is misspelled.

Line 70:

It should be "and" rather than a comma after the Hattermann and Chatterjee citations.

Line 156:

It should be "north **of** their usual locations..."

Line 193:

The expression "anomalous temperature anomaly" is unclear.

Line 224:

Northeastward is one word.

Line 226:

The last comma on this line should be removed.

Line 300:

Dall'Osto is misspelled.

**References**

Brakstad A, Våge K, Haavik L, Moore GWK. 2019. Water mass transformation in the Greenland Sea during the period 1986-2016. *Journal of Physical Oceanography* **49**: 121–140, doi:10.1175/JPO–D–17–0273.1.

Chafik L, Rossby T. 2019. Volume, heat, and freshwater divergences in the Subpolar North Atlantic suggest the Nordic Seas as key to the state of the Meridional Overturning Circulation. *Geophysical Research Letters* **46**: doi:10.1029/2019GL082 110.

Eldevik T, Nilsen JEØ, Iovino D, Olsson KA, Sandø AB, Drange H. 2009. Observed sources and variability of Nordic Seas overflow. *Nature Geoscience* **2**: 406–410, doi:10.1038/NGEO518.

Håvik L, Pickart RS, Våge K, Thurnherr AM, Beszczynska-Möller A, Walczowski W, von Appen WJ. 2017. Evolution of the East Greenland Current from Fram Strait to Denmark Strait: Synoptic measurements from summer 2012. *Journal of Geophysical Research: Oceans* : doi:10.1002/2016JC012 228.

Latarius K, Quadfasel D. 2016. Water mass transformation in the deep basins of the Nordic Seas: Analyses of heat and freshwater budgets. *Deep Sea Research I* **114**: 23–42, doi:10.1016/j.dsr.2016.04.012.

Lauvset SK, Brakstad A, Våge K, Olsen A, Jeansson E, Mork KA. 2018. Continued warming, salinifiction and oxygenation of the Greenland Sea gyre. *Tellus A* **70**: doi:10.1080/16000 870.2018.1476 434.

Mauritzen C. 1996. Production of dense overflow waters feeding the North Atlantic across the Greenland-Scotland Ridge. Part 1: Evidence for a revised circulation scheme. *Deep Sea Research I* **43**: 769–806.

Moore GWK, Våge K, Pickart RS, Renfrew IA. 2015. Open-ocean convection becoming less intense in the Greenland and Iceland Seas. *Nature Climate Change* **5**: doi:10.1038/nclimate2688.

Onarheim IH, Eldevik TE, Smedsrud LH, Stroeve JC. 2018. Seasonal and regional manifestation of Arctic sea ice loss. *Journal of Climate* **31**: 4917–4932, doi:10.1175/JCLI–D–17–0427.1.

Rogers JC, Hung MP. 2008. The Odden ice feature of the Greenland Sea and its association with atmospheric pressure, wind, and surface flux variability from reanalyses. *Geophysical Research Letters* **35**: L08 504, doi:10.1029/2007GL032 938.

Ronski S, Budéus G. 2005. Time series of winter convection in the Greenland Sea. *Journal of Geophysical Research* **110**: C04 015, doi:10.1029/2004JC002 318.

Woodgate RA, Fahrbach E, Rohardt G. 1999. Structure and transports of the East Greenland Current at 75°N from moored current meters. *Journal of Geophysical Research* **104**: 18 059–18 072.

---

## Referee Comment (RC2) · Anonymous Referee #2 · 24 Sep 2020

Review of the manuscript Ấń The impact of atmospheric and oceanic circulations on the Greenland Sea ice concentrationÂż, submitted to The Cryosphere

General comments This is a novel study that focuses on the dual influence and control that oceanic and atmospheric circulation have on several key parameters in the Greenland Sea. It aims to show how the complete system works together to shape the sea ice conditions, ocean stratification and upper ocean heat content, also aiming to explain why the characteristic sea ice shape 'Odden' tend to occur in some winters and not others. The latter through influence of Atlantic Water higher up in the water column and increased inflow of Atlantic Water during periods with an anomalous strong

gyre. The paper largely succeeds in showing this interplay. The authors also show how their results fit into a bigger picture of processes through framing it well into published literature.

One of the strengths of the paper is the focus on both vertical and horizontal processes, both from the atmosphere to the ocean and sea ice, and from the ocean to the sea ice (the impact further from the ocean/sea ice on the atmosphere is less mentioned). The paper has a high and complex aim, trying to reveal a complex interplay, and it therefore needs to be very well written and have a tidy structure in order to give a clear presentation of the results. Unfortunately, this is not good enough in the current version. Sentences are mostly well written, but the organization of the paper needs improvement, and some parts are weak or even lacking, e.g. Ch. 1 lacks a clearly stated objective and Ch. 2 lacks a thorough description of the methods. Ch. 3 Results and Discussions appear messy since sentences that belong to the introduction and methods are blended in, and there are some repetitions in this chapter. This tends to preclude the key results and make it harder to follow the discussion. However, with a largely improved presentation of the findings, including a clearer focus on the objective, an addition of a thorough description of the methods and a model evaluation, the paper will be worthy of publication. Note that even if the numerical model has been evaluated previously, this is not equivalent to it being adequate for investigating the processes which are analysed here. A model evaluation is therefore needed in this paper, see comment below. It is also important to treat uncertainties better in the paper, to show that the results are significant.

My conclusion is that an improved version of the manuscript will be well-worthy of publication and deepen the insight on how the atmosphere and ocean act in tandem in the Greenland Sea. The paper gives rise to an improved understanding of the complex and important interplay that takes place between the ocean, sea ice and atmosphere, and the results can have value for understanding regions outside the Greenland Sea as well. I encourage the authors to add a discussion on how the complete set of results

can be viewed schematically. My suggestion is a divergence/convergence situation or strong divergence (pos. GI periods) versus weak divergence (neg. GI periods) situation, see major point below.

Specific comments - It would improve the paper to include a discussion on the overall picture of processes towards the end (discussion), accompanied by a schematic illustration of the process described. What may be the overall picture of processes here? Is it strong divergence versus weak divergence with the associated Ekman transport and pumping? The analysis compares periods with a strong and weak gyre circulation and show that it corresponds to periods with sea ice transport towards the Greenland coast versus periods with sea ice transport towards the gyre and shaping of the characteristic Odden sea ice shape. Is this also valid for a larger domain encompassing the study area? I.E., is this comparable to a large divergence situation where low sea level pressure induces stronger cyclonic motion in the gyre, which in turn is associated with stronger Ekman drift of sea ice and surface water away from the gyre, inducing a lift of the interface between the fresher waters in the upper ocean and Atlantic Water below? And that this also involves Ekman pumping in the centre of the gyre, lifting the interface and inducing stronger inflow of Atlantic Water towards the gyre? When the gyre is more relaxed, there is less divergence and the sea ice can drift also towards the gyre region, particularly with the Jan Mayen Current and shape the characteristic 'Odden' tongue of sea ice. There is a need to summarize this at the end of Ch. 3 and to add schematics of the horizontal and vertical conceptual framework that the authors mention several times in the manuscript.

-Why use the smaller region to the west (72-75N, 18-10W) when investigating the co-variability between stratification strength and gyre strength in Fig. 7? These two study regions are not similar. Why not use the same region as for the Gyre Index? It is logical to use the 72-75N, 18-10W for showing flow of AW upstream of the gyre, but for the stratification comparison I assume it is better to look at the actual response of the gyre. From reading the manuscript one gets the impression that the figure intends

to show the change in stratification in the gyre itself, not upstream. -Present the Gyre Index thoroughly early on, as it is a key part of the analysis. This can be done by adding a time series of it in Fig. 1b and highlight the positive and negative time periods that are used for the analysis. A rationale for using the chosen threshold values 0.75 and $-0.75$ is needed (in the methods). The paper needs to show which time periods are used in the analysis, as the negative and positive periods are compared. Explain in the methods carefully how you have estimated the "composite differences". This explanation can refer to Fig. 1b, the time series of the Gyre Index. It can also be of good value to present maps showing the mean wind fields for the positive and negative periods as Fig 1c and d or add such maps to Fig. 4.

-The last paragraph of Ch. 1 needs a strong rewrite. Clearly state the objective of the paper and mention briefly in one or two sentences how the study is performed, which data have been used and how the rest of the paper is structured/organized. In the current version, the objective and hypothesis are mentioned indirectly in Ch. 3. -The Method section is poor. Ch. 2 Data and methods presents the data shortly but lacks a thorough description of the methods. Reorganize it to e.g. 2.1 Data, with three paragraphs on atmospheric, oceanic and sea ice data, and 2.2 Methods, with a careful explanation of what the authors did in this paper. An evaluation of the TOPAZ4 results are needed to show that the model results are appropriate for investigating the objective of the paper. Mention which oceanographic data are used for the data assimilation (how many, from which data sets and the distribution seasonally and through the time period). What are the typical discrepancies between the TOPAZ4 before and after the data assimilation? What are the parts of the ocean-sea ice system that the model performs well on and which is it not simulating well?

-The evaluation needs to show that the TOPAZ4 results simulate reasonably well the key variables investigated in this study, which include the ocean stratification, sea ice concentration and ocean circulation in the upper 500 m., in terms of the spatial pattern of the mean fields and their temporal variability. It is also important to show the vertical

structure of the ocean in temperature and salinity to check that TOPAZ4 reproduces the change in these variables between positive and negative Gyre Index periods. Add a brief discussion on TOPAZ4's applicability, strengths and weaknesses that shows why the model is suitability for the purpose here. Even if there are discrepancies from reality in the model results, they can still be useful for the purpose here. However, the discrepancies need to be shown and mentioned explicitly and it is necessary to discuss the findings in consideration of these discrepancies at the end of the paper. E.G., from Fig. 2 is clear that TOPAZ4 has a higher sea ice concentration variability than the observations. Does this imply that it exaggerates the variability, which in the time series analysis could result in a higher significance than in reality?

-Ch. 3 Results and Discussion is messy and needs a strong tidying job. Having the results and discussion in parallel may work, but that demands a very well written, tidy and organized results and discussion chapter. As it is reads now, Ch. 3 is a mixture of results and discussion blended with introduction and methods sentences. Stick to a structure of starting each paragraph with briefly stating a result and referring to the companying figure. Then discuss briefly what the finding means and how it is interpreted by the authors and where applicable shortly if that is in line or not with published literature. The paper needs a broader, separate discussion of the findings before the conclusions are drawn. This can be a last paragraph of Ch. 3 that presents the flowchart in Fig. 8 and a new schematic in Fig. 8b that shows the process in the horizontal and vertical.

-Conceptual presentation: For this type of paper, that encompasses a complex and overarching picture of processes it is important to present the conceptual framework explicitly. Fig. 8 needs to be strengthened with a schematic of how the concept it is viewed from above, showing the case with the strong gyre/positive GI periods, with arrows showing the divergence of sea ice (and surface water) and stronger inflow of Atlantic Water, and a vertical sketch showing the lifted/raised pycnocline, with AW higher up in the water column. Then explain that during neg. Gyre Index periods, the gyre is

more relaxed, there is less divergence/less Ekman transport and sea ice tends to be drifting more towards the gyre and the 'Odden' tongue of sea ice can be formed (helped by the JMC). In this way, the author's interpretation of the whole set of processes can be summarized and made easily accessible for readers and increase the impact of the paper.

-Seasonal aspects: The paper does the analyses for winter (Dec-Jan-Feb), presumably because winter has strong wind forcing, thus likely a stronger signal-to-noise ratio which increases the chances of tracing the signals under investigation, with significant correlation coefficients. This is a reasonable choice, but the authors need to provide a rationale for it and discuss if the investigated processes are in effect and significant in other seasons. It is necessary that the model simulations are good in winter. I assume there is less observational CTD data to assimilate the model with in winter. It is therefore necessary to show the seasonal distribution of the observational data and that the model performs well enough in winter (particularly, the change between winters with different forcing is essential here).

-It has not been shown that AW reaches the surface in response to the anomalously strong gyre circulation. In the analysis, this is checked for the upper 400 m and even if the mean temperature and salinity increases in the upper 400 m, this is not equivalent to showing that AW reaches the surface. Either modify the wording to e.g. "implying a lift of the AW" or "raising the AW" or similar or strengthen the analysis to show that it does occur. It will strengthen the paper if the analysis is made for more specific parts of the water column, e.g. for 0-100 m and 100-400 m separately, or other depth spans that the authors find more appropriate for investigating how AW is lifted in the water column in response to the strengthened gyre. Fig. 7 is informative in this manner, and should be expanded to include Hovmöller diagrams of temperature and salinity in the same way as panels b and c.

-There is a poor level of treatment of uncertainties in the manuscript. There are only a few years that goes into the statistical analysis (i.e. those with Gyre Index >0.75

or $< -0.75$) and it is important to show that the results are significant. Estimate the uncertainty and show it, in time series as e.g shading around each variable and in maps as shading or hatching over the significant areas. Write the p-value after each correlation coefficient. Why use the 95 % confidence level? Are the results valid on the 99 % confidence level?

Technical corrections TITLE -Remove the period at the end. -Consider a rewrite to highlight the finding, i.e. that there is a combined impact by the atmosphere and ocean. And since the ocean is the main focus here, consider mentioning it first, thus changing to "Combined impact of oceanic and atmospheric circulations on Greenland Sea ice conditions". -Since the authors investigate the effect not only on sea ice concentration but also on ocean stratification and AW inflow, the title could end with "...Greenland Sea conditions".

ABSTRACT -Regards the sentence "This in turn decreases the freshwater content and weakens the ocean stratification in the central GS". This may very well be true, but it has not been properly shown that the freshwater content declined in response to a stronger Gyre Index and the associated Ekman transport of sea ice and freshwater away from the gyre. Can you add figures to show this? Is the freshwater input reduced in response to less sea ice transported into the GS? But the effect of that would be seen after the following summer? Second last sentence: It has not been shown that AW is reaching the surface. It has been shown that there are warmer waters in the upper 400 m when the gyre is strong (GI>0.75). And how high up does the authors mean when they use the phrase "surface"? Upper 400 m is very large span and increased heat content in the upper 400 m probably reflect that the Atlantic Water is occupying more of the water column when the pycnocline is raised more (during periods with stronger divergence and increased Ekman transport of surface waters and sea ice away from the gyre).

CH. 1 INTRODUCTION Paragraph 40-45: -Last sentence: 'can also be important in terms of interactions' is unclear. Rephrase. Paragraph 50-55: -Second sentence:

rephrase 'or the old sea ice' to 'or older sea ice'. The sentence reads as though there is either young or old ice. Can younger and older sea ice occur in the Odden region at the same time? Paragraph 60-65: -First sentence: change to 'large-scale'. -Third sentence. Rewrite and add a hyphen between 'NAO' and 'like'. Suggesting a rewrite to 'The large-scale atmospheric circulation resembles the pattern of the North Atlantic Circulation (NAO), but the NAO-like pattern is not covarying significantly with the Odden ice extent (Comiso et al. 2001).' Paragraph 70-75: -First sentence: Remove comma after parenthesis with references and add 'and'. -Sentence starting with 'Further, . . .' Add em-dash around the embedded clause: 'Further, the eastward flowing JMC—originated from the EGC—constitutes the . . .' -Second last sentence: Change 'this' to 'the' in 'this cold and fresh JMC'. -Last sentence: Remove 'current' in 'JMC current'. Also, note that this sentence mentions indirectly the hypothesis of the paper. Rather write it explicitly (e.g. starting with 'Our hypothesis is. . .') or rewrite the sentence to e.g. 'This implies that the GSG circulation . . .' and add a sentence stating the hypothesis explicitly in the following paragraph, where also the objective needs to be clearly formulated. Paragraph 80-85 – last paragraph of the introduction: A clear ending of the introduction is needed. This is an important paragraph of the paper and needs to state explicitly the objective of the paper, the hypothesis, which now is mentioned indirectly in the paragraph above and in sentences in Ch. 3. Also mention briefly how aim is investigated and how the paper is structured. -First sentence: Be specific and explicit. Add what is investigated before stating the aim, e.g. 'In this study, we investigate . . . with the aim to . . .' -Add a sentence between the current first and second sentences stating the hypothesis: 'Our hypothesis is that . . .'. -Second sentence: Add briefly the approach of comparing time periods with weak and strong forcing, e.g. 'Using a combination of . . ., we compare time periods with strong and weak gyre circulation and show that . . .' -Last sentence: 'Further, it is shown that . . .' is a rather vague and passive start of the sentence. Rewrite to e.g. 'We also show that . . .'. Consider to rewrite '. . . helps setting up . . .' to something more clear.

CH. 2 DATA AND METHODS Paragraph 100: -This is a too short introduction to the

atmospheric data. There is information missing. State all the atmospheric variables that were used in the study. Were they monthly means? What more did you do with the atmospheric data? What were they used for? This is completely missing from Ch. 2. -Use en-dash, not hyphen when stating the time period, i.e., '1991–2017'. -add a 'the' in front of 'ERA'. -Use past tense in the data and methods chapter, consistently. Paragraph 105-110: -Add which vertical resolution there is in the model simulation. This is important in order for it to be useful for studying the change in vertical structure of the water column and the changes in stratification between positive and negative Gyre Index periods. -Last sentence: add a comma after 'observations'. Remove 's' in 'temperatures' and 'sea ice concentrations'. -From where were the observations collected? Which data set? How many profiles, and from which years, seasons, etc. -Add an evaluation of the TOPAZ4. Is the model simulation suitable for the purpose here? This needs to be shown. Paragraph 115-120: -Rewrite to 'Following Chatterjee et al. (2018), we estimated the strength of the GSG circulation by area-averaging the winter-mean December-January-February (DJF) barotropic stream function within the 3000 m isobath in the region 73–78 °N, 12 °W–9 °E (Fig. 1).' Which depth span was used in the water column for this estimation? Add the 3000 m isobath as a thicker contour in Fig. 1 so that it is easily seen which contour this is referring to. -Second sentence: Rewrite to 'The area-averaged value was standardized over the complete time period 1991–2017 to . . .'. The phrase 'to get the' is a bit awkward. Consider refining it. Also, it would fit well to refer to a Fig. 1b with the time series of the Gyre Index here, as mentioned above. -Third sentence: A rationale is needed for choosing the thresholds 0.75 and −0.75. -Last sentence: Is this only for the oceanic data? This shows that it is tidier to have methods as a separate subchapter, not mention methods within the 2.2. Oceanic data. Paragraph in Ch. 2.3: Again, this is too brief. Add how the data was used in the analysis. -First sentence: Rewrite to 'Monthly mean sea ice concentration data . . . were obtained from . . .'. At which grid size? -Second sentence: Rewrite to 'Sea ice velocity data were obtained from . . .' *METHODS section is missing. What did you do here? Add a proper evaluation of TOPAZ4. Explain the methods, mentioning

the time periods, study area, that you used winter-mean values (and why), that you used anomalies, etc. How many years actually went into the analysis. Highlight in bands with shading in new Fig. 1b so that it can be seen how many winters (DJF) had a positive Gyre Index > 0.75, and how many was <−0.75. This will allow for a higher transparency and clarity through the paper.

CH. 3 RESULTS AND DISCUSSION Use past tense when presenting the results. Use present tense when discussing them. Start each paragraph with presenting a result and refer to the appropriate figure. Then discuss what the results can imply towards the end of the paragraph. End Ch. 3 with a separate paragraph with a broader discussion of the results and the authors interpretation of them. Start this paragraph with presenting Fig. 8 (and add a conceptual sketch to it). Paragraph 125-130: -The first three sentences belong to Ch. 1. The rest of the paragraph belongs to the evaluation. I suggest moving them to Ch. 2. -Forth sentence: Move to evaluation part in Ch. 2. Refine to "The standard deviation of winter-mean DJF SIC shows high variability along the MIZ and the Odden region, in the observational and reanalysis data (Fig. 2)." -Last sentence: Move sentence to evaluation part in Ch. 2. Also, add that the figure shows that TOPAZ4 has a higher interannual variability in the winter-mean DJF sea ice concentration compared with the observations. Add how this discrepancy can or will influence the results and conclusions of this study. Is it exaggerating the interannual variability, thus giving a higher signal to noise-ratio, or is the additional variability not in phase with the observed variability? Please add a time series of the corresponding SIC variability of the observational and reanalysis SIC in the study area as a Fig. 2c. The model was assimilated with SIC data. Mention why it still has discrepancies from the observed SIC data. How often is the model assimilated (daily, monthly?). Paragraph 140-145: This paragraph is messy, it contains parts of the objective, results, methods and discussion, and it suffers from some poor, vague writing. It needs a strong rewrite, and the sentences that remain in this paragraph needs to be sharp "to the point". -First sentence is about the objective of the paper. Move it to the last paragraph of Ch. 1 and rewrite to a clear sentence. -Sentences 2-4 belong to this paragraph as they present the result and

the accompanying brief discussion of the implication of the result, which is appropriate for the chapter. -Fifth and sixth sentences belongs to Ch. 1. I understand that the authors have a need to explain why they move on to investigating the atmospheric fields, but there is too much introductory text here, 'with full sentences simply stating the results of other references. Rather simply add a '..., in line with *REFS*' after the fourth sentence. -The last two sentences are a mixture of methods and discussion. I suggest taking them out and, if necessary, starting the following paragraph with a brief mentioning of why the atmospheric influence was investigated. Paragraph 155-175: Very long paragraph, which presents two different figures. Split into two paragraphs where Fig. 5 is presented for the first time. -The first sentence is unnecessary long and difficult to read. Rephrase it. The part "suggests that the large-scale circulation associated with the GSG circulation features a NAO-like meridional pattern although the SLP..." could be written as "shows a NAO-like atmospheric pattern associated with the GSG ocean circulation, but with centres of action north of their usual locations (Fig. 4a). " Further, it is not self-explanatory what is meant with the phrase "composite differences of anomalies of ". I understand from reading the paper that you have estimated the difference in winter SLP between the positive and negative Gyre Index periods, using anomalies from the long-term mean, but this should be explicitly written in the methods chapter and also understandable just from reading the figure caption and introduction of the figure in the Ch. 3. If you want to keep the phrase "composite differences of SLP anomalies" then explain its meaning explicitly in the methods. Write which time-period the anomalies are estimated based on, in the text and in the caption. -Sentence starting with "The GSG circulation responds to...": I suggest bringing it further up front in the paragraph, as the second sentence. Then say that the correlation coefficient with the static, traditional NAO-pattern is insignificant, showing the importance of taking spatial variability of NAO's centres of action into account. -The two sentences starting with "However, at the same time...": It is awkward and needs a rewrite, to e.g. "The associated wind stress can influence sea ice transport through Ekman transport, towards the central GS and Greenlands west-coast due Ekman transport" and "We find

anomalous northerly wind stress in the central GS during the positive Gyre Index periods, and vice versa for southerly wind stress." Comment: Is it a southerly wind stress or a weaker northly wind stress during negative Gyre Index periods? It would be clarifying to show the maps with the mean wind fields for the two different cases, strong and weak gyre. -Panels are introduced in the opposite sequence. Introduce Fig. 5a first, then 5b. -This last part around lines 170-175 contains key discussion that is important and may fit better in a separate paragraph for the broader discussion at the end of Ch. 3. Paragraph 185-190: -The entire paragraph is introductory text and includes what the paper investigates in the third sentence. Move this information to Ch. 1. Paragraph 195-200: -First and second sentences: Rewrite to state the result explicitly, then end with '(Fig. 6a)'. The term 'composite differences' is not very intuitive. Could this be rephrased to something more easily understood? The point is that the figure shows that the ocean temperature in the upper 400 m is higher during the positive Gyre Index periods, right? Please write this simply and explicitly and mention how much higher the temperature is during these periods compared with the mean of the negative Gyre Index periods. -Third sentence: Remove. This is repetition and another mentioning of introductory text that does not belong to Ch. 3. -Fourth sentence: Rewrite to start with bluntly presenting the result coming from Fig. 6b, i.e. 'There is a significant positive correlation between the Gyre Index and ocean heat transport in the upper 400 m in the smaller study area west of the main study area (r=0.7)'. (The phrase will improve when the two study areas are named and introduced in the same panel in Fig. 1a.) It is cluttering the structure to start with how a previous finding is confirmed here. This is not a proper way to introduce Fig. 6b. Again, stick to the pattern of first presenting the result, and secondly discuss briefly its meaning, implication, mention how it confirms a previous finding or the like. -Last sentence: If this is also for the upper 400 m then it mixes the signatures of variability in the AW with the upper water masses. It would be more appropriate to check separately for different parts of the water column, e.g. 0-100 m and 100-400 m, separately, as AW is typically in the latter whereas the surface waters and polar/Arctic waters occupies the upper part. That would allow for

seeing if the surface water decrease in salinity in periods with negative Gyre Index and more sea ice drift towards the gyre and would give a stronger result on whether AW is influencing higher up in the water column during positive Gyre Index periods. The term 'surfacing' is too strong given that it is likely rising but not necessarily reaching the surface. Be specific about which region this is estimated for. Paragraph 210-215: The paragraph is messy and needs to be tidied up. It starts with discussing figures 4 and 5. Rather start it with 'The Gyre Index is covarying with the Brunt-Väisälä frequency (Fig. 7).' And add which depth span of the water column the stratification is estimated over. I suggest continuing with 'Our comparison shows that a weakening of the stratification in the upper part of the water column coincides with a stronger GSG circulation and vice versa.' -Sentence 4: Rewrite to 'This supports that . . . by the GSG can rise under a . . ., hence potentially also the SIC.' -Last sentence: Delete 'further' and 'eastward flowing'. Change 'EGC' to 'JMC'. Delete the clause starting with ', which constitutes' because it is repetition. *A summarizing paragraph with a broader discussion is needed here. This should start with presenting Fig. 8 and write up the complete picture of processes and how the authors interpret their findings.

CH. 4 CONCLUSIONS Please state if atmospheric or oceanic circulation when 'circulation' is mentioned in the conclusion, to avoid confusion. -First sentence: Unclear. Please rewrite. I suggest starting with e.g. 'Here, we investigate . . . and show that . . .'. -Second sentence: add 'the' before 'wind stress curl'. -Third sentence: Rewrite to be more specific, e.g. 'The large-scale atmospheric circulation pattern that influences the GSG circulation resembles a NAO-like pattern with its northern centre of action situated northeast of the typical NAO pattern.' -Fourth sentence: Add a 's' in 'sea ice conditions'. After 'Odden region', add 'in the GS'. Modify end of sentence to 'through Ekman drift of sea ice toward the Greenland coast during periods with northerly winds (Germe et al. 2011).' -Fifth sentence: The sentence is a bit messy. Rewrite it to e.g. 'During periods with anomalously low SLP and strong gyre circulation in the GS, northerly winds and associated Ekman drift causes sea ice drift towards the Greenland coast. This reduces the SIC in the central GS.' -Sixth sentence: Consider a rewrite to 'We show

that this is associated with a weakening of the stratification in the upper water column.' or similar, to be more direct and specific. -Seventh sentence: Add a comma after 'into the central GS'. Modify the latter part of the sentence. It has not been shown that the AW reaches all the way to the surface, only that the upper 400 m become warmer and less saline. -Eight sentence: Presentation of Figure 8 is too short and should have been mentioned at the end of Ch. 3 and with a broader discussion of the conceptual view of the findings. -Last sentences after "...(Fig. 8)": This part is taking up too much space in the conclusion and gives the impression of dampening the value of the findings of the paper and the closure of the paper becomes too vague. Rather mention this clearly in the methods, that the impact of other smaller scale processes would largely cancel out or act to reduce the correlation coefficients in the processes studied here. But despite these smaller scale processes the results are significant. However, if mentioned in the conclusion it could be rephrased to e.g. "Despite the presence of smaller scale processes, such as eddies and wave interactions, our results on the larger scale processes are significant with high correlation coefficients. This implies that smaller scale processes largely cancel out over time or are not strong enough to dampen the larger scale processes, at least not when comparing periods with weak and strong gyre circulation in winter when the wind forcing is strong".

FIGURES In general, figure captions are lacking information and are not complete, and there are incomplete sentences, e.g. when explaining place names in Fig 1. Make sure all the information is given for each figure. The introduction of study areas should be made up front in Figure 1. In the current version of the manuscript the reader is asked to check for other figures to see which study region is meant. Rather include the two study regions and bathymetry in all the maps were those are needed in order to interpret the results from that figure. Be consistent. Consider to move the larger map from Fig. 4 to Fig. 1, as it fits with zooming in to the study area, and can be referred to in Ch. 1 Introduction. The term "composite differences" is used without further explanation, but is not self explanatory. Please be clear so the reader does not have to check the methods to understand what the figure shows. It can be written out

full without too much space, e.g. 'Difference in winter-mean SLP for DJF between time periods with strong and weak Gyre Index during 1991-2017', or something similar.

Figure 1: -Consider moving the larger map in Fig. 4a to here to zoom in early on. -Add the other smaller study region as well and name the two. -It is a bit misleading to show the larger study area as a box when in reality is following a bathymetry contour. Rather show it with the 3000 m bathymetry contour. Consider showing it in all the figures that have a map. It should be possible without cluttering the figures. -Add a time series of the Gyre Index as Fig. 1b, and highlight the positive and negative time periods and threshold values 0.75 and −0.75 as e.g. shaded bands and dotted lines. -Highlight the 3000 m isobath contour as e.g. a thicker contour line than the others, to show the region where the Gyre Index is estimated from.

Figure 2: -This figure belongs to the Methods section, showing the variability of the sea ice concentration fields in winter and is used for evaluation of TOPAZ4. Rather refer to it in Ch. 2. -Add label to the colorbar. -Rephrase 'winter (DJF) mean' to 'winter-mean DJF'. -Add more panels to evaluate the model thoroughly. It is important to show that it simulated the ocean stratification and temperature and salinity well.

Figure 3: -Caption: Explanation of the red square is missing. -This can also be part of the evaluation and mentioned for the first time in Ch. 2. Why is the co-variability between SIC and the Gyre Index stronger in TOPAZ4 compared with the observations? How can this influence the results and the interpretation of them? This should be discussed in the methods section. -Fig. 3b: Panel is denoted as (a). -Interpretation of Fig. 3: The authors conclude on causality when the figure only shows inverse co-variability. It could be that SIC and the Gyre Index are both affected by the atmospheric wind forcing? Sentence in paragraph 140-145 could be rephrased to e.g.: "This indicate that the GS SIC variability is covarying with the GSG circulation." -Correct the first sentence of the caption to '... (a) satellite observations and (b) the TOPAZ4 reanalysis...'.

Figure 4: -Show also the mean wind fields of positive and negative Gyre Index periods,

not only the anomalies to show the differences between the positive and negative periods. Is the mean of the negative periods a weaker northerly wind field compared to the mean of the positive periods, or is it a southerly wind field, with wind from the south? I assume it is not a mean southerly wind field in the negative periods, but a weaker northerly wind field, and that the anomalies show southerly wind because they are less northerly compared with the temporal mean for the whole study period. But this needs to made clear for the reader. Showing the mean fields would make the interpretations more intuitive and the paper easier to follow. -In the caption, add which time period the anomalies are estimated from. Add which data set the SLP is from. -Consider to add similar maps as b and c for Ekman pumping to make a stronger argument for the "lift of AW" during positive Gyre Index periods. -This figure has different coastlines compared with the other maps. Please be consistent.

Figure 5: -In caption, delete 'vectors'. -Again, showing only the anomalies means it is not entirely clear if the positive Gyre Index periods are associated with weaker southward sea ice drift or if the mean sea ice drift is from the other direction, i.e., northward sea ice drift. It would be more intuitive and easier to follow the manuscript if the mean field is shown for sea ice drift in positive and negative Gyre Index periods.

Figure 6: -Panel 6a: Add outlines for the study areas from which the variables in 6b are estimated from. -Panel 6b: Add line for y=0. Add bands of shading showing the positive and negative Gyre Index periods, from which the map in 6a is estimated from. -Consider adding a similar map as that of panel 6a for the upper 100 m or even the upper 50 m. This could be very interesting and give information that helps make the interpretation of the complete picture of processes (i.e., regards Ekman transport of surface water). -First sentence in caption is unclear. Rephrase to 'Difference in average potential temperature anomalies in the upper 400 m of the water column between positive and negative Gyre Index periods during 1991–2017.' -The term 'temperature advection' is perhaps better phrased as 'heat transport'. Explain in the methods how it was estimated. -Second sentence in caption is unclear. Rephrase to 'Time series of

the Gyre Index (blue curve) and standardized anomalies of the salinity and temperature advection in the upper 400 m.' Please do not use the term 'surface salinity' for the salinity in the upper 400 m, as 'surface' is typically associated with the upper 0-50 m or so.

Figure 7: -Add similar panels for temperate and salinity in ocean and increase the depth span to show the lift of AW. This can help justify the conclusions regards the lifting of AW in periods with positive Gyre Index. -Why is the Brunt-Väisälä frequency not estimated for the same region as the Gyre Index? Using the other smaller region outside the gyre makes the interpretation harder. The main response regards the lifting of the interface between the upper polar water masses and AW is occurring most strongly in the centre of the gyre?

Figure 8: -Flip the diagram on the side, with the atmospheric pathway on top and the oceanic pathway below. -Add schematics of how the authors interpret the process in the horizontal (showing divergence of sea ice and freshwater due to Ekman transport in response to stronger wind forcing and related increased AW recirculation and inflow to the GS) and vertically (showing the Ekman pumping and lift of the AW in response).

USAGE OF TERMS -use the term "winter-mean (DJF)" instead of "winter time (DJF)". -Avoid abbreviations as much as possible for easy reading. -the term "northerly" is used for both wind direction and sea ice drift. To avoid confusion, consider using "northerly" and "southerly" only for wind, and "northward" for sea ice drift and oceanic currents. See e.g. second sentence at the beginning of page 8, where the usage of "northerly" for both wind and sea ice drift is confusing. It is not clear if the sea ice drift is from the north or from the south from this sentence. -use apostrophes only when introducing a new term, like 'the Gyre Index'. Then refer to it simply as the Gyre Index without apostrophes on later mentions. The same goes for the Odden region. -Present the two study regions in the methods section, new Ch. 2.2., in Fig. 1a and Fig. 1 captions. Name the two study regions and use these names consistently throughout the paper. Write in the methods what you have estimated for each study area. -Which depth range

is the surface salinity anomaly estimated for? Write this in the methods. If this is the upper 400 m then "upper ocean salinity" is more appropriate. -The term 'validation' is used in section 2.2. 'Evaluation' is a better suited term because it reflects that all models have strengths and weaknesses, no models are perfect, and a key point is to make sure that the model results are useful for investigating the objective of the paper with the chosen approach.

---

## Author Comment (AC1) · 27 Oct 2020

**Response to Reviewer's comments (R1) on**

The impact of atmospheric and oceanic circulations on the Greenland Sea ice concentration
by Sourav Chatterjee, Roshin P. Raj, Laurent Bertino, Sebastian H. Mernild, Nuncio Murukesh, and
Muthalagu Ravichandran

Reviewer's Comments:

In this manuscript the interannual variability of sea ice concentration in the Greenland Sea is
investigated. The authors identify several atmospheric and oceanic processes that influence the sea
ice concentration, and clarify how these are modulated by the large-scale atmospheric conditions.
The authors conclude that the magnitude of the Greenland Sea Gyre circulation is of particular
importance.

I think this is an interesting manuscript that highlights the importance of changing sea ice
concentrations in the Greenland Sea. My main concern is that the changes in sea ice and ocean
conditions over the period considered (1991-2017) are more appropriately characterized by secular
trends than interannual variability, in particular reduced sea ice concentration and a warming ocean.
I think the variability investigated in this manuscript needs to be discussed within the context of
these long-term trends. As such, I recommend that the paper be revised before publication.

Authors' reply:

We thank the reviewer for the spending valuable time for going through the manuscript and
providing constructive comments and references for improving the manuscropt. A point by point
response to the reviewer comments is listed below.

Major comment:

Sea ice concentration in the western Nordic Seas has steadily diminished over the past decades
(Moore et al., 2015; Onarheim et al., 2018). In particular, the Odden ice tongue has rarely formed
since the 1990s (e.g. Rogers and Hung, 2008). Over the same period equally remarkable changes in
stratification and water mass transformation in the Greenland Sea have taken place (Ronski and
Budéus, 2005; Latarius and Quadfasel, 2016; Lauvset et al., 2018; Brakstad et al., 2019). Yet these
secular trends, which dominate the variability investigated in the manuscript, are barely, if at all,
mentioned. This is important context that needs to be discussed and accounted for.

We thank the reviewer for raising this important issue. Indeed, the Greenland Sea water mass and
the Sea ice concentration in the western Nordic Seas have been transforming with secular trend. We
will incorporate these aspects in the revised manuscript, which will surely help to improve the
manuscript.

Also please note that, our main objective is to find the process(es)/mechanisms through which
Greenland Sea gyre (GSG) affects the sea ice concentration of the region. In the revised version we
have shifted the focus to the western Greenland Sea, where the interanual variation and the effect of
GSG is most prominent (Figure 2 and 3) instead of the 'Odden' region. The effect of the large scale
GSG circulation on the 'Odden' formation is not very clear from our study. This may be due to the
fact that the occasional 'Odden' formation depends on various small scale processes such as waves,
eddies etc. The introduction is revised to incorporate the new changes.

Also note that to minimize the effect of the trends on the interannual variability all our statistical analysis we have performed with detrended data.

Specific comments:

Line 29:
The first two sentences of the introduction are too strong. As the source of dense overflow waters that supply the deep limb of the Atlantic Meridional Overturning Circulation, the Nordic Seas are indeed very important (e.g. Chafik and Rossby, 2019). But for the Greenland Sea to control regional and hemispheric climate, the Greenland Sea would have to be a main source of overflow water. This is likely not the case (Mauritzen, 1996; Eldevik et al., 2009).

Yes we agree. We will modify this part accordingly.

Lines 37-45:
Please clarify that the Odden ice feature rarely developed after the 1990s (e.g. Rogers and Hung, 2008) and that in the present climate only intermediate (not deep) waters are formed in the Greenland Sea (e.g. Brakstad et al., 2019).

We thank the reviewer for the valuable comment. We will discuss about it in the revised manuscript. Please note that, our main objective is to expalin the mechanisms through which Greenland Sea gyre circulation effects the sea ice. Since the Odden has been rarely developed during the time period considered in our study, we focus to the western Greenland Sea where we find the effect of GSG circulation on sea ice is most prominent.

Line 57:
Please clarify where the high sea level pressure anomaly patter would have to be located in order to result in anomalous southerly wind in the Greenland Sea.

Noted. Will be clarified in the revised version of the manuscript.

Line 70:
Please clarify how the Greenland Sea Gyre contributes to heat distribution in the Nordic Seas. It seems more plausible that heat inside a gyre would be trapped rather than distributed.

We will expand this discussion with more details. The main idea here is to highlight that, on the eastern side of the Nordic Seas GSG helps in propagation of northward flowing Atlantic water towards Fram Strait (Chatterjee et al., 2018), on its western side it brings the recirculated Atlantic water from Fram Strait to Greenalnd Sea region (Hatterman et al., 2016).

Line 74:
Please also clarify to what extent a strengthened western branch of the Greenland Sea Gyre circulation results in increasing Atlantic Water transport into the central Greenland Sea vs. Atlantic Water throughput as part of the East Greenland Current (Woodgate et al., 1999). The bulk of the Atlantic Water remains within the East Greenland Current and is transported toward Denmark Strait (Håvik et al., 2017).

Thank you for suggesting the point. The recirculation of Atlantic water (AW) with the GSG circulation has been reported by Hatterman et al. (2016). The warming signals along the GSG pathway (Figure 6) clearly shows the influence of GSG on warm AW transport in Greenland Sea.

Indeed, the strengthening of gyre through low SLP can strengthen the East Greenland Current (EGC), however, we don't see a clear warming along the EGC in Figure 6. Although in general the bulk of the Atlantic Water remains within the East Greenland Current and is transported toward Denmark Strait (Håvik et al., 2017), in a strong GSG condition, its contribution to AW recirculation can increase than normal. And also note that this AW recirculated by GSG, is warmer and thus important for sea ice.

Line 108:
Has TOPAZ been evaluated against observations in the central Greenland Sea? Latarius and Quadfasel (2016) or Brakstad et al. (2019) would be good points of comparison.

Evaluation of TOPAZ against observations will be added to the revised version of the manuscript.

Line 142:
Does the regression map show significant negative sea ice concentration in the central Greenland Sea when the Greenland Sea Gyre is strong?

It shows significant sea ice concentration pattern in the western Greenland Sea. Our current definition of 'central Greenland Sea' partly includes this region. We will modify the definitions of the regions to bring more clarity.

The regression map is shown in Figure 3, both with observation and TOPAZ and they show comparable patterns. In the map only significant values are shown. The region of our interest with maximum influence is marked.

Line 155:
Please clarify that it (presumably) is the large-scale atmospheric circulation associated with the Greenland Sea Gyre circulation that features an NAO-like pattern.

Yes we agree. The points will be clarified in the revised version of the manuscript. Figure 4a shows the SLP anomaly pattern effectively during the strong GSG periods, which is similar to NAO.

Line 159:
It is unclear how the low correlation between the gyre and NAO indices signifies an importance of NAO on the circulation in the Greenland Sea.

Here we tried to highlight that it signifies the importance of the **spatial variability** of NAO. Note that the NAO-like pattern associated with GSG (Figure 4a) has its centre north of its usual locations. The low correlation between GSG and NAO index (reflective of its usual spatial pattern with an Icelandic low) thus highlights the significance of the location of the SLP minimum in the Nordic Seas.

We agree that the above mentioned points are not clear and will be made clearer in the revised version.

Line 161:
Please clarify how winds influence the drift of sea ice. Is the drift primarily determined by Ekman transport or directly by the wind? To what extent does that depend on sea ice concentration?

We will clarify the point mentioned by the reviewer in the revised version of the manuscript. The sea ice in the Greenland Sea is either formed locally or exported through the Fram Strait. In the

latter case, it is heavily deformed and drifts almost freely under the actions of the winds and surface currents. This is even more true at ice concentrations lower than 80%.

Line 172:
The statement that wintertime Greenland Sea sea ice concentration and Fram Strait ice are flux are not strongly correlated appears to directly contradict the statement on line 35 that changes in ice export through Fram Strait influence the Greenland Sea sea ice concentration.

We will correct it.

Line 188:
Does the gyre bring Atlantic Water into the central Greenland Sea or circulate Atlantic Water around the periphery of the Greenland Sea?

We will concentrate more on the terminology. Central Greenland sea (at the core of the gyre) is not where we intend to focus for sea ice changes. The reason is mentioned on the second comment. We show that the AW causes sea ice changes in the western Greenland Sea.

Our results suggest that the gyre assist AW around the periphery of the Greenland Sea that inturn cause sea ice changes in the western Greenland Sea. We don't have clear evidence to claim the same in the central Greenland Sea (and/or Odden). This is made clear in the revised manuscript.

Line 230:
Please expand on how Atlantic Water anomalies would impact sea ice formation and how that may influence convection.

As suggested by the reviewer the discussion on the impact of AW anomalies on sea ice formation will be expanded. Furthermore new analysis will be included to show the relation more clearly.

Line 251:
The central Greenland Sea has largely been ice free since the 1990s (Moore et al., 2015; Brakstad et al., 2019). This large-scale sea ice retreat, consistent with sea ice loss across the entire Arctic region, is likely not related to the magnitude of the Greenland Sea Gyre circulation.

We agree and thank the reviewer for bringing our attention to this very important topic. Please note that the revised version of the manuscript focuses mainly on western Greenland Sea region. We, in the revised version, will highlight the role of GSG in bringing warm water to the western Greenland Sea, thus impacting the sea ice of the region.

Figure 1:
The black oval indicating the central Greenland Sea extends onto the Greenland shelf, which should not be considered part of the central Greenland Sea.

Thank you for pointing this out. The figure will be modified accordingly.

Figure 2:
Is it reasonable to average sea ice concentration over the entire 1991-2017 period if the variability is dominated by an ice Odden "on" or "off" state? To the extent that the Odden feature is binary (on or

off), the average would represent an in-between state that is never realized. Perhaps consider comparing instead observations and TOPAZ for years when Odden is present and for years when it is not.

In Figure 2, the standard deviation of the winter mean (DJF) for the study period is shown. Thus, one may expect that it is the binary feature of the Odden which is depicted in the figure. This will be made clear in the revised version of the manuscript.

Figure 6:
What are the correlations between salinity anomaly, temperature advection, and gyre index? Please clarify in the caption what temperature advection means. Is it heat transport or a product of temperature and velocity, and where is it evaluated?

The correlations between both salinity anomaly and temperature advection with Gyre index is 0.7 and is mentioned in the text. It is product of temperature and velocity in the marked box in Figure 3. This will be made clear in the revised version of the manuscript.

Figure 7:
It appears that one buoyancy frequency profile per year is shown in Fig. 7. Are the values annual means or summertime means? Please clarify. For most of the year the mixed-layer depth in the Greenland Sea is deeper than 50 m and the buoyancy frequency would be very low. I think it would be sensible to consider the stratification to a deeper level in the Greenland Sea. These days buoyancy frequency seems to be more commonly used than Brunt-Väisälä frequency.

The values are winter (DJF) means and clarified in the text. As per suggestion, the depth will be taken till 100 m in the new figure.

Detailed comments:
Line 32:
It should be "... from the central Arctic Ocean ..."
Will be Corrected

Line 59:
Although is misspelled.
Will be corrected

Line 70:
It should be "and" rather than a comma after the Hattermann and Chatterjee citations.
Will be corrected

Line 156:
It should be "north of their usual locations..."
Will be corrected

Line 193:
The expression "anomalous temperature anomaly" is unclear.
Will be corrected

Line 224:
Northeastward is one word.
Will be corrected

Line 226:
The last comma on this line should be removed.
Will be corrected

Line 300:
Dall'Osto is misspelled.
Will be corrected.

---

## Author Response (AR1)

**Response to Reviewer's comments (R1) on**

The impact of atmospheric and oceanic circulations on the Greenland Sea ice concentration
by Sourav Chatterjee, Roshin P. Raj, Laurent Bertino, Sebastian H. Mernild, Nuncio Murukesh, and
Muthalagu Ravichandran

Reviewer's Comments:

In this manuscript the interannual variability of sea ice concentration in the Greenland Sea is
investigated. The authors identify several atmospheric and oceanic processes that influence the sea
ice concentration, and clarify how these are modulated by the large-scale atmospheric conditions.
The authors conclude that the magnitude of the Greenland Sea Gyre circulation is of particular
importance.

I think this is an interesting manuscript that highlights the importance of changing sea ice
concentrations in the Greenland Sea. My main concern is that the changes in sea ice and ocean
conditions over the period considered (1991-2017) are more appropriately characterized by secular
trends than interannual variability, in particular reduced sea ice concentration and a warming ocean.
I think the variability investigated in this manuscript needs to be discussed within the context of
these long-term trends. As such, I recommend that the paper be revised before publication.

Authors' reply:

We thank the reviewer for the spending valuable time for going through the manuscript and
providing constructive comments and references for improving the manuscropt. A point by point
response to the reviewer comments is listed below.

Major comment:

Sea ice concentration in the western Nordic Seas has steadily diminished over the past decades
(Moore et al., 2015; Onarheim et al., 2018). In particular, the Odden ice tongue has rarely formed
since the 1990s (e.g. Rogers and Hung, 2008). Over the same period equally remarkable changes in
stratification and water mass transformation in the Greenland Sea have taken place (Ronski and
Budéus, 2005; Latarius and Quadfasel, 2016; Lauvset et al., 2018; Brakstad et al., 2019). Yet these
secular trends, which dominate the variability investigated in the manuscript, are barely, if at all,
mentioned. This is important context that needs to be discussed and accounted for.

We thank the reviewer for raising this important issue. Indeed, the Greenland Sea water mass and
the Sea ice concentration in the western Nordic Seas have been transforming with secular trend. We
have incorporated these aspects in the revised manuscript.

Please note that, our main objective is to find the process(es)/mechanism(s) through which
Greenland Sea gyre (GSG) affects the sea ice concentration of the region. In the revised version we
have shifted the focus primarily to the western Greenland Sea, where the interanual variation and
the effect of GSG is most prominent (Figure 2 and 5), instead of the 'Odden' region. The effect of
the large scale GSG circulation on the 'Odden' formation is not very clear from our study possibly
due to rare occurences of the Odden in recent years. But in western GS the response of the GSG
circulation to the atmospheric forcing and its consequences are quite noticable. Although studying
the changes in watermass in central GS is not the main objective here, we have discussed toward
the end of the manuscript about the observed changes in watermasses in central GS and how the
response of GSG circulation to the atmospheric forcing can be associated with that

Also note that to minimize the effect of the trends on the interannual variability all our statistical analysis we have performed with detrended data.

Specific comments:

Line 29:
The first two sentences of the introduction are too strong. As the source of dense overflow waters that supply the deep limb of the Atlantic Meridional Overturning Circulation, the Nordic Seas are indeed very important (e.g. Chafik and Rossby, 2019). But for the Greenland Sea to control regional and hemispheric climate, the Greenland Sea would have to be a main source of overflow water. This is likely not the case (Mauritzen, 1996; Eldevik et al., 2009).

In the revised manuscript, we have removed the part on Greenland Sea's role on regional and hemispheric climate.

Lines 37-45:
Please clarify that the Odden ice feature rarely developed after the 1990s (e.g. Rogers and Hung, 2008) and that in the present climate only intermediate (not deep) waters are formed in the Greenland Sea (e.g. Brakstad et al., 2019).

We thank the reviewer for the valuable comment. In the revised manuscript the points have been mentioned in Line # 40-42 . Please note that, our main objective is to expalin the mechanisms through which Greenland Sea gyre circulation effects the sea ice. Since the Odden has been rarely developed during the time period considered in our study, we focus to the western Greenland Sea where we find the effect of GSG circulation on sea ice is most prominent.

Line 57:
Please clarify where the high sea level pressure anomaly patter would have to be located in order to result in anomalous southerly wind in the Greenland Sea.

Noted. We have modified the sentence in line no. 50-51: *For example, a high sea level pressure (SLP) anomaly over the NS results in anomalous southerly wind in the GS.*

Line 70:
Please clarify how the Greenland Sea Gyre contributes to heat distribution in the Nordic Seas. It seems more plausible that heat inside a gyre would be trapped rather than distributed.

We have expanded this with more details. The main idea here is to highlight that, on the eastern side of the Nordic Seas GSG helps in propagation of northward flowing Atlantic water towards Fram Strait (Chatterjee et al., 2018), on its western side it brings the recirculated Atlantic water from Fram Strait to Greenalnd Sea region (Hatterman et al., 2016). In the revised manuscript it has been explained in line nos. 57-63 as below:
*The Greenland Sea Gyre (GSG) is a prominent feature of the subpolar North Atlantic ocean and can be intensified as a strong cyclonic circulation in the NS (Fig. 1). It is known to respond to the atmospheric forcing in the NS and contribute to AW heat distribution in the Nordic Seas (Hatterman et al. 2016; Chatterjee et al. 2018). A stronger GSG circulation increases the AW temperature in the FS by modifying the northward AW transport in its eastern side (Chatterjee et al. 2018). Simultaneous increase in its southward flowing western branch, constituting the southern recirculation pathway of AW (Hattermann et al. 2016; Jeansson et al. 2017), increases the heat content in the western GS through a stronger and warmer recirculation of AW (Chatterjee et al. 2018).*

Line 74:
Please also clarify to what extent a strengthened western branch of the Greenland Sea Gyre circulation results in increasing Atlantic Water transport into the central Greenland Sea vs. Atlantic Water throughput as part of the East Greenland Current (Woodgate et al., 1999). The bulk of the Atlantic Water remains within the East Greenland Current and is transported toward Denmark Strait (Håvik et al., 2017).

Thank you for suggesting the point. The recirculation of Atlantic water (AW) with the GSG circulation has been reported by Hatterman et al. (2016). The warming signals along the GSG pathway (Figure 8a) clearly shows the influence of GSG on warm AW transport in Greenland Sea. Indeed, the strengthening of gyre through low SLP can strengthen the East Greenland Current (EGC), however, we don't see a clear warming along the EGC in Figure 8a. Although in general the bulk of the Atlantic Water remains within the East Greenland Current and is transported toward Denmark Strait (Håvik et al., 2017), in a strong GSG condition, its contribution to AW recirculation can increase than normal. And also note that this AW recirculated by GSG, is warmer and thus important for sea ice, if they can come to surface.

Line 108:
Has TOPAZ been evaluated against observations in the central Greenland Sea? Latarius and Quadfasel (2016) or Brakstad et al. (2019) would be good points of comparison.
We have a separate section (section 3) for evaluation of TOPAZ4 in the revised manuscript. Note that as explained in above comments, since we are interested on the sea ice changes in the western GS and oceanic changes responsible for that, we have chosen an area for evaluation of TOPAZ4 in the western GS where the sea ice concentration variability and also GSG's influence are maximum. We use EN4 observation for the evaluation.

Line 142:
Does the regression map show significant negative sea ice concentration in the central Greenland Sea when the Greenland Sea Gyre is strong?

It shows significant sea ice concentration pattern in the western Greenland Sea. Our current definition of 'central Greenland Sea' partly includes this region. We will modify the definitions of the regions to bring more clarity.

The regression map is shown in Figure 5, both with observation and TOPAZ4 and they show comparable patterns. In the map only significant values are shown. The region of our interest with maximum influence is marked.

Line 155:
Please clarify that it (presumably) is the large-scale atmospheric circulation associated with the Greenland Sea Gyre circulation that features an NAO-like pattern.

In the revised manuscript we have discussed about this in line nos. 187-196.

*To elucidate the possible influence of atmospheric circulation pattern associated with GSG circulation on the SIC variability in the GS, linear regression of the sea level pressure anomalies on the gyre index was calculated and shown in Fig. 6. The large-scale atmospheric circulation shows a positive NAO-like pattern associated with a strong GSG circulation, but with centres of actions north of their usual locations (Fig. 6). The GSG circulation responds to the anomalous wind stress curl induced by the low SLP anomaly patterns in the NS (Chatterjee et al. 2018). However, we found that the station based NAO index, with its spatial feature highlighting the Icelandic low and Azores high, (https:// climatedataguide.ucar.edu/sites/default/files/nao_station_seasonal.txt) and the gyre index have a very low correlation (r = 0.2). This further points to the importance of the spatial variability of NAO (Zhang et al. 2008; Moore et al. 2012)*

*and its influence on the Nordic Seas circulation. Also note that the low correlation could rise from the fact that the equatorward pole of NAO doesn't exhibit much significant regression patterns in Fig. 6.*

Line 159:

It is unclear how the low correlation between the gyre and NAO indices signifies an importance of NAO on the circulation in the Greenland Sea.

Here we tried to highlight that it signifies the importance of the **spatial variability** of NAO. Note that the NAO-like pattern associated with GSG (Figure 6) has its centre north of its usual locations. The low correlation between GSG and NAO index (reflective of its usual spatial pattern with an Icelandic low) thus highlights the significance of the location of the SLP minimum in the Nordic Seas.

We have included a detailed discussion . Please see Line nos. 187-196 (or italic lines in the above comment) in the revised manuscript.

Line 161:
Please clarify how winds influence the drift of sea ice. Is the drift primarily determined by Ekman transport or directly by the wind? To what extent does that depend on sea ice concentration?

The sea ice in the Greenland Sea is either formed locally or exported through the Fram Strait. In the latter case, it is heavily deformed and drifts almost freely under the actions of the winds and surface currents. This is even more true at ice concentrations lower than 80%. However note that, the ekman transport effect on the seaice is discussed in Germe et al., (2011) and also our analysis showing a reduced southward sea ice flow in the Greenalnd Sea interior (Fig. 7) even in presence of a low SLP (northerly winds) over the GS justifies the presence of ekman transport of sea ice towards the Greenland coast.
Line 172:
The statement that wintertime Greenland Sea sea ice concentration and Fram Strait ice are flux are not strongly correlated appears to directly contradict the statement on line 35 that changes in ice export through Fram Strait influence the Greenland Sea sea ice concentration.

Please note that, while the ice export from FS is an important factor for determining the GS SIC, the variability of sea ice concentration in the GS can be largely determined by the local meteorological and oceanic conditions which may weaken the correlation between Greenland Sea ice concentration and Fram Strait ice area flux. In fact, Selyuzhenok et al. (2020) found that in spite of increasing sea ice export through the FS, the overall sea ice volume (SIV) in the GS has been decreasing during the period 1979–2016. This is largely due to changes in local oceanic control on sea ice.

Line 188:
Does the gyre bring Atlantic Water into the central Greenland Sea or circulate Atlantic Water around the periphery of the Greenland Sea?

We will concentrate more on the terminology. Central Greenland sea (at the core of the gyre) is not where we intend to focus for sea ice changes. The reason is mentioned on the second comment. We show that the AW causes sea ice changes in the western Greenland Sea.

Our results suggest that the gyre assist AW around the periphery of the Greenland Sea that inturn cause sea ice changes in the western Greenland Sea. We don't have clear evidence to claim the same in the central Greenland Sea (and/or Odden).

Line 230:
Please expand on how Atlantic Water anomalies would impact sea ice formation and how that may influence convection.

As suggested by the reviewer the discussion on the impact of AW anomalies on sea ice formation is expanded in Line nos. 228-235 and 258-268 . Furthermore new analysis (Figure 9) is included to show the relation more clearly. Kindly note that the convection takes place mostly in the core of the gyre. However we are focusing on the sea ice impacts outside the gyre. So we have not discussed about convection in the Results section. Although a discussion on it is included towards the end in lines: 292-300

Line 251:
The central Greenland Sea has largely been ice free since the 1990s (Moore et al., 2015; Brakstad et al., 2019). This large-scale sea ice retreat, consistent with sea ice loss across the entire Arctic region, is likely not related to the magnitude of the Greenland Sea Gyre circulation.

We agree and thank the reviewer for bringing our attention to this very important topic. Please note that the revised version of the manuscript focuses mainly on western Greenland Sea region. We, in the revised version, have highlighted the role of GSG in bringing warm water to the western Greenland Sea, thus impacting the sea ice of the region.

Note that, Atlantification of the Nordic Seas is known to be a reason large scale sea ice retreat in other parts of the Arctic Ocean (Polyakov et al., 2017; Lind et al., 2018). Here we further show that this is the case even in Greenland Sea and the asoociation of atlantification and sea ice in GS is constituted by the gyre circulation.

Figure 1:
The black oval indicating the central Greenland Sea extends onto the Greenland shelf, which should not be considered part of the central Greenland Sea.

Thank you for pointing this out. The figure is modified accordingly.

Figure 2:
Is it reasonable to average sea ice concentration over the entire 1991-2017 period if the variability is dominated by an ice Odden "on" or "off" state? To the extent that the Odden feature is binary (on or off), the average would represent an in-between state that is never realized. Perhaps consider comparing instead observations and TOPAZ for years when Odden is present and for years when it is not.

In Figure 2, the standard deviation of the winter mean (DJF) for the study period is shown. Thus, one may expect that it is the binary feature of the Odden which is depicted in the figure. However, our focus is on the western GS and not on the Odden sea ice variability.

Figure 6:
What are the correlations between salinity anomaly, temperature advection, and gyre index? Please clarify in the caption what temperature advection means. Is it heat transport or a product of temperature and velocity, and where is it evaluated?

It correspondes to Figure 8 in the revised manuscript. The correlations between both salinity anomaly and temperature advection with Gyre index is 0.7 and is mentioned in the text. It is determined as $U.\nabla T$ in the marked box in Figure 8a. It is mentioned in the figure caption now.

Figure 7:
It appears that one buoyancy frequency profile per year is shown in Fig. 7. Are the values annual means or summertime means? Please clarify. For most of the year the mixed-layer depth in the Greenland Sea is deeper than 50 m and the buoyancy frequency would be very low. I think it would be sensible to consider the stratification to a deeper level in the Greenland Sea. These days buoyancy frequency seems to be more commonly used than Brunt-Väisälä frequency.

The values are winter (DJF) means and clarified in the text. As per suggestion, the depth will be taken till 100 m in the new figure.

Detailed comments:
Line 32:
It should be "... from the central Arctic Ocean ..."
Corrected

Line 59:
Although is misspelled.
Removed

Line 70:
It should be "and" rather than a comma after the Hattermann and Chatterjee citations.
Removed

Line 156:
It should be "north of their usual locations..."
corrected

Line 193:
The expression "anomalous temperature anomaly" is unclear.
Modified

Line 224:
Northeastward is one word.
Text modified

Line 226:
The last comma on this line should be removed.
Text modified

Line 300:
Dall'Osto is misspelled.
Corrected.

The impact of atmospheric and oceanic circulations on the Greenland Sea ice concentration
by Sourav Chatterjee, Roshin P. Raj, Laurent Bertino, Sebastian H. Mernild, Nuncio Murukesh, and Muthalagu Ravichandran

Reviewer's Comments:

Review of the manuscript The impact of atmospheric and oceanic circulations on the Greenland Sea ice concentration, submitted to The Cryosphere General comments This is a novel study that focuses on the dual influence and control that oceanic and atmospheric circulation have on several key parameters in the Greenland Sea. It aims to show how the complete system works together to shape the sea ice conditions, ocean stratification and upper ocean heat content, also aiming to explain why the characteristic sea ice shape 'Odden' tend to occur in some winters and not others. The latter through influence of Atlantic Water higher up in the water column and increased inflow of Atlantic Water during periods with an anomalous strong gyre. The paper largely succeeds in showing this interplay. The authors also show how their results fit into a bigger picture of processes through framing it well into published literature.

One of the strengths of the paper is the focus on both vertical and horizontal processes, both from the atmosphere to the ocean and sea ice, and from the ocean to the sea ice (the impact further from the ocean/sea ice on the atmosphere is less mentioned). The paper has a high and complex aim, trying to reveal a complex interplay, and it therefore needs to be very well written and have a tidy structure in order to give a clear presentation of the results. Unfortunately, this is not good enough in the current version. Sentences are mostly well written, but the organization of the paper needs improvement, and some parts are weak or even lacking, e.g. Ch. 1 lacks a clearly stated objective and Ch. 2 lacks a thorough description of the methods. Ch. 3 Results and Discussions appear messy since sentences that belong to the introduction and methods are blended in, and there are some repetitions in this chapter. This tends to preclude the key results and make it harder to follow the discussion. However, with a largely improved presentation of the findings, including a clearer focus on the objective, an addition of a thorough description of the methods and a model evaluation, the paper will be worthy of publication. Note that even if the numerical model has been evaluated previously, this is not equivalent to it being adequate for investigating the processes which are analysed here. A model evaluation is therefore needed in this paper, see comment below. It is also important to treat uncertainties better in the paper, to show that the results are significant.

My conclusion is that an improved version of the manuscript will be well-worthy of publication and deepen the insight on how the atmosphere and ocean act in tandem in the Greenland Sea. The paper gives rise to an improved understanding of the complex and important interplay that takes place between the ocean, sea ice and atmosphere, and the results can have value for understanding regions outside the Greenland Sea as well. I encourage the authors to add a discussion on how the complete set

of results can be viewed schematically. My suggestion is a divergence/convergence situation or strong divergence (pos. GI periods) versus weak divergence (neg. GI periods) situation, see major point below.

Authors' reply:

The authors thank the reviewer and highly acknowledge the effort for such in detail evaluation and valuable suggestions which will strengthen the study. We will work on the presentation of the results and make it more clear to follow.

Specific comments

It would improve the paper to include a discussion on the overall picture of processes towards the end (discussion), accompanied by a schematic illustration of the process described. What may be the overall picture of processes here? Is it strong divergence versus weak divergence with the associated Ekman transport and pumping? The analysis compares periods with a strong and weak gyre circulation and show that it corresponds to periods with sea ice transport towards the Greenland coast versus periods with sea ice transport towards the gyre and shaping of the characteristic Odden sea ice shape. Is this also valid for a larger domain encompassing the study area? I.E., is this comparable to a large divergence situation where low sea level pressure induces stronger cyclonic motion in the gyre, which in turn is associated with stronger Ekman drift of sea ice and surface water away from the gyre, inducing a lift of the interface between the fresher waters in the upper ocean and Atlantic Water below? And that this also involves Ekman pumping in the centre of the gyre, lifting the interface and inducing stronger inflow of Atlantic Water towards the gyre? When the gyre is more relaxed, there is less divergence and the sea ice can drift also towards the gyre region, particularly with the Jan Mayen Current and shape the characteristic 'Odden' tongue of sea ice. There is a need to summarize this at the end of Ch. 3 and to add schematics of the horizontal and vertical conceptual framework that the authors mention several times in the manuscript.

Thank you for suggesting these important points. As suggested a schematic picture detailing the overall process is added as Figure 10 in the revised manuscript.

The divergence related with gyre circulation can also be important along with the wind driven Ekman transport. Thank you for the useful suggestion. This is incorporated in the text. Please see section 5 and Figure 10.

Also please note that, our main objective is to find the process(es)/mechanisms through which Greenland Sea gyre (GSG) affects the sea ice concentration of the region. In the revised version we have particularly focused on the western Greenland Sea, where the interanual variation and the effect of GSG is most prominent (Figure 2 and 5) instead of the 'Odden' region (and/or centre of the gyre). Note that since the Odden has formed very rarely in recent years since 1990s (e.g. Rogers and Hung, 2008), we do not get significant response of sea ice concentration in Odden region to the gyre circulation.

Why use the smaller region to the west (72-75N, 18-10W) when investigating the covariability between stratification strength and gyre strength in Fig. 7? These two study regions are not similar. Why not use the same region as for the Gyre Index? It is logical to use the 72-75N, 18-10W for showing flow of AW upstream of the gyre, but for the stratification comparison I assume it is better to look at the actual response of the gyre. From reading the manuscript one gets the impression that the figure intends to show the change in stratification in the gyre itself, not upstream.

The influence of gyre on sea ice is shown Fig 5. Note that the marked region is where (1) the influence of gyre index on sea ice is maximum and also the same region has (2) the strongest interannual variability (Fig 2). Also in the core of the gyre and/or in the Odden region, sea ice is ocassionaly formed, making it difficult to meet our objective i.e role of atmosphere ocean dynamics on interannual variability of sea ice. So for the strength of gyre circulation, the core of the gyre is chosen, but for its impact on sea ice, the western Greenland Sea region with the two features above are chosen. We have mentioned these points in the revised manuscript in line nos. 144-147.

Present the Gyre Index thoroughly early on, as it is a key part of the analysis. This can be done by adding a time series of it in Fig. 1b and highlight the positive and negative time periods that are used for the analysis. A rationale for using the chosen threshold values 0.75 and −0.75 is needed (in the methods). The paper needs to show which time periods are used in the analysis, as the negative and positive periods are compared. Explain in the methods carefully how you have estimated the "composite differences". This explanation can refer to Fig. 1b, the time series of the Gyre Index. It can also be of good value to present maps showing the mean wind fields for the positive and negative periods as Fig 1c and d or add such maps to Fig. 4.

As suggested the Gyre index has been introduced at the beginning as Fig 1c. The strong and weak gyre index periods are marked separately in Figure 8b and accompanied with results from composite analysis for readers' convinience. The 0.75 threshold was chosen to consider only the sufficiently strong/weak gyre circulation periods (Line 123-124). In the method section (Line 121-123) the composite analysis is described in and also explained in Figure 8 caption. The composite analysis for the SLP is removed.

The last paragraph of Ch. 1 needs a strong rewrite. Clearly state the objective of the paper and mention briefly in one or two sentences how the study is performed, which data have been used and how the rest of the paper is structured/organized. In the current version, the objective and hypothesis are mentioned indirectly in Ch. 3.

The paragraph is rewritten (Line 73-79)
*In this study we hypothesize that the interannual winter mean SIC variability in GS can be explained by the combined influence of atmospheric and oceanic circulations, more precisely the GSG circulation. Using a combination of satellite*

*passive microwave SIC, a coupled sea ice ocean reanalysis and atmospheric reanalysis data, we show that changes in the GSG dynamics and resulting AW transport in GS can potentially influence the SIC in the western GS. Further, we also show that the atmospheric circulation associated with the GSG circulation variability provides the favourable conditions for the GSG's control on the SIC variability in the western GS region. Section 2 and 3 describe the data and methods applied in the study following the results in section 4. Discussions and conclusions are mentioned in section 5.*

The Method section is poor. Ch. 2 Data and methods presents the data shortly but lacks a thorough description of the methods. Reorganize it to e.g. 2.1 Data, with three paragraphs on atmospheric, oceanic and sea ice data, and 2.2 Methods, with a careful explanation of what the authors did in this paper. An evaluation of the TOPAZ4 results are needed to show that the model results are appropriate for investigating the objective of the paper. Mention which oceanographic data are used for the data assimilation (how many, from which data sets and the distribution seasonally and through the time period). What are the typical discrepancies between the TOPAZ4 before and after the data assimilation? What are the parts of the ocean-sea ice system that the model performs well on and which is it not simulating well?

Thank you for pointing this out. We have rewritten the data nad methoda section with more details.
 Note that the detailed setup and performance of the TOPAZ4 reanalysis is exposed in Xie et al. 2017, including the counts of observations and the temporal variations of the data counts. Of particular relevance for the Greenland Sea are the assimilation of Argo profiles, research cruises CTDs from IOPAS and AWI (Sakov et al. 2012), satellite sea ice concentration, sea surface temperatures and sea level anomalies. This is mentioned in section 2.2. The changes in performance in TOPAZ4 after data assimailation is described in Lien et al. (2016).

The evaluation needs to show that the TOPAZ4 results simulate reasonably well the key variables investigated in this study, which include the ocean stratification, sea ice concentration and ocean circulation in the upper 500 m., in terms of the spatial pattern of the mean fields and their temporal variability. It is also important to show the vertical structure of the ocean in temperature and salinity to check that TOPAZ4 reproduces the change in these variables between positive and negative Gyre Index periods. Add a brief discussion on TOPAZ4's applicability, strengths and weaknesses that shows why the model is suitability for the purpose here. Even if there are discrepancies from reality in the model results, they can still be useful for the purpose here. However, the discrepancies need to be shown and mentioned explicitly and it is necessary to discuss the findings in consideration of these discrepancies at the end of the paper. E.G., from Fig. 2 is clear that TOPAZ4 has a higher sea ice concentration variability than the observations. Does this imply that it exaggerates the variability, which in the time series analysis could result in a higher significance than in reality?

We have added analysis for evaluation of TOPAZ4 by comaring with EN4 observations (Figs. 3 and 4). Section 2.4 expalins the results from the comparison. Note that, since we are interested in role of oceanic (and atmospheric) processes in influecing the sea ice in the western GS, we have compared

TOPAZ4 and EN4 in the western GS only. Comparisons are shown for surface and AW layers separately. A discussion on the results from the comparisons is given in Lines 143-161.

Results and Discussion is messy and needs a strong tidying job. Having the results and discussion in parallel may work, but that demands a very well written, tidy and organized results and discussion chapter. As it is reads now, Ch. 3 is a mixture of results and discussion blended with introduction and methods sentences. Stick to a structure of starting each paragraph with briefly stating a result and referring to the companying figure. Then discuss briefly what the finding means and how it is interpreted by the authors and where applicable shortly if that is in line or not with published literature. The paper needs a broader, separate discussion of the findings before the conclusions are drawn. This can be a last paragraph of Ch. 3 that presents the flowchart in Fig. 8 and a new schematic in Fig. 8b that shows the process in the horizontal and vertical.

We thank the reviewer for the detailed constructive comment. We have now results and discussions as separate sections with the suggested schematic of the processes studied in the study.

Conceptual presentation: For this type of paper, that encompasses a complex and overarching picture of processes it is important to present the conceptual framework explicitly. Fig. 8 needs to be strengthened with a schematic of how the concept it is viewed from above, showing the case with the strong gyre/positive GI periods, with arrows showing the divergence of sea ice (and surface water) and stronger inflow of Atlantic Water, and a vertical sketch showing the lifted/raised pycnocline, with AW higher up in the water column. Then explain that during neg. Gyre Index periods, the gyre is more relaxed, there is less divergence/less Ekman transport and sea ice tends to be drifting more towards the gyre and the 'Odden' tongue of sea ice can be formed (helped by the JMC). In this way, the author's interpretation of the whole set of processes can be summarized and made easily accessible for readers and increase the impact of the paper.

We agree with the suggestion! A schematic of the explained proceeses that influences the western GS sea ice is summarized and added in the discussions.

Seasonal aspects: The paper does the analyses for winter (Dec-Jan-Feb), presumably because winter has strong wind forcing, thus likely a stronger signal-to-noise ratio which increases the chances of tracing the signals under investigation, with significant correlation coefficients. This is a reasonable choice, but the authors need to provide a rationale for it and discuss if the investigated processes are in effect and significant in other seasons. It is necessary that the model simulations are good in winter. I assume there is less observational CTD data to assimilate the model with in winter. It is therefore necessary to show the seasonal distribution of the observational data and that the model performs well enough in winter (particularly, the change between winters with different forcing is essential here).

Our rationale is the fact that the sea ice in the region are mostly present during winter only. Thus all though the processes may still be active during other seasons, but to show their impact on sea ice we need to choose winter months only. Also note that the key candidate influencing western GS sea ice, the GSG circulation, also found to be strongest during winter (Figure 1c). This is mentioned in lines 119-121.

Kindly note that, the quality assessment of TOPAZ4 has been explored in Xie et al (2017). From the findings of that study it is noted that the errors in sea ice concentrations in the Greenland Sea are smaller in winter than summer due to the stronger thermal forcing by the atmosphere (Fig. 13 in their paper). Also, Fig. 7 (right) in their paper suggests SST performs overall better in winter than in summer due to the mixed layer dynamics. Also regarding observations over Greenland Sea (GS), note that, GS is poorly observed (or observations were not available publicly for assimilation) all year round even accumulating observations during their study period (1991-2013). In either case, their study did not notice that the seasonality of assimilated profiles influenced the performance of the TOPAZ4 reanalysis system. We believe This may become a more prominent issue once more profiles become publicly available.

It has not been shown that AW reaches the surface in response to the anomalously strong gyre circulation. In the analysis, this is checked for the upper 400 m and even if the mean temperature and salinity increases in the upper 400 m, this is not equivalent to showing that AW reaches the surface. Either modify the wording to e.g. "implying a lift of the AW" or "raising the AW" or similar or strengthen the analysis to show that it does occur. It will strengthen the paper if the analysis is made for more specific parts of the water column, e.g. for 0-100 m and 100-400 m separately, or other depth spans that the authors find more appropriate for investigating how AW is lifted in the water column in response to the strengthened gyre. Fig. 7 is informative in this manner, and should be expanded to include Hovmöller diagrams of temperature and salinity in the same way as panels b and c.

As suggested Hovmöller diagrams of temeprature and salinity is included to show the vertical mixing of AW and increase in upper ocean temperature and salinity with stronger gyre circulation. They are shown as Figure 9b,c in the revised manuscript.

There is a poor level of treatment of uncertainties in the manuscript. There are only a few years that goes into the statistical analysis (i.e. those with Gyre Index >0.75 or <−0.75) and it is important to show that the results are significant. Estimate the uncertainty and show it, in time series as e.g shading around each variable and in maps as shading or hatching over the significant areas. Write the p-value after each correlation coefficient. Why use the 95 % confidence level? Are the results valid on the 99 % confidence level?

The composite analysis with strong and weak gyre periods are used for potential temperature and freshwater content difference between those periods and shown in Fig.8 and 11 respectively, with significant areas marked. Correlation values, where used (line 218,234), are accompanied with p-values. All results are valid at 99% except the results from the composite analysis in Fig 8 and 11, which is significant at 95%. For consistency throughout we have used 95% level.

Technical corrections

TITLE -Remove the period at the end. -Consider a rewrite to highlight the finding, i.e. that there is a combined impact by the atmosphere and ocean. And since the ocean is the main focus here, consider mentioning it first, thus changing to "Combined impact of oceanic and atmospheric circulations on Greenland Sea ice conditions". -Since the authors investigate the effect not only on sea ice concentration but also on ocean stratification and AW inflow, the title could end with ". . .Greenland Sea conditions".

We insist on changing the title as "Combined impact of oceanic and atmospheric circulations on Greenland Sea ice concentrations" for reasons: 1) we are only looking at sea ice concentration and not any other parameters e.g thickness, area, volume. So ending with "sea ice conditions" may not be appropiate. 2) Further, the changes in startification and AW inflow are the reasons which ultimately is shown to end up with affecting sea ice. They may influence other properties as well e.g bottom water formation which we have not touched here as our main objective is to highlight the impact on sea ice concentrations.

ABSTRACT -Regards the sentence "This in turn decreases the freshwater content and weakens the ocean stratification in the central GS". This may very well be true, but it has not been properly shown that the freshwater content declined in response to a stronger Gyre Index and the associated Ekman transport of sea ice and freshwater away from the gyre. Can you add figures to show this? Is the freshwater input reduced in response to less sea ice transported into the GS? But the effect of that would be seen after the following summer? Second last sentence: It has not been shown that AW is reaching the surface. It has been shown that there are warmer waters in the upper 400m when the gyre is strong (GI>0.75). And how high up does the authors mean when they use the phrase "surface"? Upper 400 m is very large span and increased heat content in the upper 400 m probably reflect that the Atlantic Water is occupying more of the water column when the pycnocline is raised more (during periods with stronger divergence and increased Ekman transport of surface waters and sea ice away from the gyre).

We have added analysis of freshwater content as Figure 11. Along with reduced sea ice export, the northerly winds and the stronger GSG circulation both can tend to push the local surface waters towards Greenland coast, freshwater content anomaly can be induced in the GS as shown in Fig 11. We have rephrased the second last sentence as *Under a weakly stratified condition, enhanced vertical mixing of these subsurface AW anomalies can warm the surface waters and inhibit new sea ice formation, further reducing the SIC in the western GS.* (Line 22-24 in the revised manuscript)

CH. 1 INTRODUCTION

Paragraph 40-45: -Last sentence: 'can also be important in terms of interactions' is unclear. Rephrase. Paragraph 50-55: -Second sentence: rephrase 'or the old sea ice' to 'or older sea ice'. The sentence reads as though there is either young or old ice. Can younger and older sea ice occur in the Odden region at the same time?

The sentence is removed in revised manucript.
Since we don't have enough signal of GSG's impact on Odden sea ice, we have reduced the discussion of Odden and focused on the region affected by the GSG.

Paragraph 60-65: -First sentence: change to 'large-scale'. -Third sentence. Rewrite and add a hyphen between 'NAO' and 'like'. Suggesting a rewrite to 'The large-scale atmospheric circulation resembles the pattern of the North Atlantic Cir- culation (NAO), but the NAO-like pattern is not covarying significantly with the Odden ice extent (Comiso et al. 2001).' Paragraph 70-75: -First sentence: Remove comma after parenthesis with references and add 'and'. -Sentence starting with 'Further, . . .'Add em-dash around the embedded clause: 'Further, the eastward flowing JMC—o-
riginated from the EGC—constitutes the . . .' -Second last sentence: Change 'this' to 'the' in 'this cold and fresh JMC'. -Last sentence: Remove 'current' in 'JMC current'. Also, note that this sentence mentions indirectly the hypothesis of the paper. Rather write it explicitly (e.g. starting with 'Our hypothesis is. . .') or rewrite the sentence to e.g. 'This implies that the GSG circulation . . .' and add a sentence stating the hypothesis explicitly in the following paragraph, where also the objective needs to be clearly formulated. Paragraph 80-85 – last paragraph of the introduction: A clear ending of the introduction is needed. This is an important paragraph of the paper and needs to state explicitly the objective of the paper, the hypothesis, which now is mentioned indirectly in the paragraph above and in sentences in Ch. 3. Also mention briefly how aim is investigated and how the paper is structured. -First sentence: Be specific and explicit. Add what is investigated before stating the aim, e.g. 'In this study, we investigate . . . with the aim to . . .' -Add a sentence between the current first and second sentences stating the hypothesis: 'Our hypothesis is that . . .'. -Second sentence: Add briefly the approach of comparing time periods with weak and strong forcing, e.g. 'Using a combination of . . ., we compare time periods with strong and weak gyre circulation and show that . . .' -Last sentence: 'Further, it is shown that . . .' is a rather vague and passive start of the sentence. Rewrite to e.g. 'We also show that . . .'. Consider to rewrite '. . . helps setting up . . .' to something more clear.

We thank the reviewer for more specific suggestions which helped to improve the readability of the manuscript. The introduction section is rewritten based on these suggestions.

CH. 2 DATA AND METHODS

Paragraph 100: -This is a too short introduction to the atmospheric data. There is information missing. State all the atmospheric variables that were used in the study. Were they monthly means? What more did you do with the atmospheric data? What were they used for? This is completely missing from Ch. 2.

We have used monthly anomaly derived from the monthly climatology for the period 1991-2017. Necessary details are incorporated in Section 2.1

 -Use en-dash, not hyphen when stating the time period, i.e., '1991–2017'. -add a 'the' in front of 'ERA'. -Use past tense in the data and methods chapter, consistently. Paragraph 105-110: -Add which vertical resolution there is in the model simulation. This is important in order for it to be useful for studying the change in vertical structure of the water column and the changes in stratification between positive and negative Gyre Index periods. -Last sentence: add a comma after 'observations'. Remove 's' in 'temperatures' and 'sea ice concentrations'. -From where were the observations collected? Which data set? How many profiles, and from which years, seasons, etc. -Add an evaluation of the TOPAZ4. Is the model simulation suitable for the purpose here? This needs to be shown.

The target densities levels are
24.05,
24.96,
25.68
26.05
26.30
26.60
26.83
27.03
27.20
27.33
27.46
27.55
27.66
27.74
27.82
27.90
27.97
28.01
28.04
28.07
28.09
28.11
28.13
and the 5 top layers are imposed z-levels, top level is 3m thick and increase by 18% each level. A separate section with relevant TOPAZ4 evaluation is added.

Paragraph 115-120: -Rewrite to 'Following Chatterjee et al. (2018), we estimated the strength of the GSG circulation by area-averaging the winter-mean December-January-February (DJF) barotropic

stream function within the 3000m isobath in the region 73–78 ◦ N, 12 ◦ W–9 ◦ E (Fig. 1).' Which depth span was used in the water column for this estimation? Add the 3000 m isobath as a thicker contour in Fig. 1 so that it is easily seen which contour this is referring to. -Second sentence: Rewrite to 'The area-averaged value was standardized over the complete time period 1991–2017 to . . .'. The phrase 'to get the' is a bit awkward. Consider refining it. Also, it would fit well to refer to a Fig. 1b with the time series of the Gyre Index here, as mentioned above. -Third sentence: A rationale is needed for choosing the thresholds 0.75 and −0.75. -Last sentence: Is this only for the oceanic data? This shows that it is tidier to have methods as a separate subchapter, not mention methods within the 2.2. Oceanic data. Paragraph in Ch. 2.3: Again, this is too brief. Add how the data was used in the analysis. -First sentence: Rewrite to 'Monthly mean sea ice concentration data . . . were obtained from . . .'. At which grid size? -Second sentence: Rewrite to 'Sea ice velocity data were obtained from . . .' *METHODS section is missing. What did you do here? Add a proper evaluation of TOPAZ4. Explain the methods, mentioning the time periods, study area, that you used winter-mean values (and why), that you used anomalies, etc. How many years actually went into the analysis. Highlight in bands with shading in new Fig. 1b so that it can be seen how many winters (DJF) had a positive Gyre Index > 0.75, and how many was <−0.75. This will allow for a higher transparency and clarity through the paper.

The whole depth was used to estimate the stream function. We have mentioned the data resolutions clearly in the data section. All other corrections suggested by the reviewer are incorporated into the revised manuscript. As asked we have added a detailed 'Data' and a seperate 'Methods' section describing all the methods applied.

CH. 3 RESULTS AND DISCUSSION Use past tense when presenting the results. Use present tense when discussing them. Start each paragraph with presenting a result and refer to the appropriate figure. Then discuss what the results can imply towards the end of the paragraph. End Ch. 3 with a separate paragraph with a broader discussion of the results and the authors interpretation of them. Start this paragraph with presenting Fig. 8 (and add a conceptual sketch to it). Paragraph 125-130: -The first three sentences belong to Ch. 1. The rest of the paragraph belongs to the evaluation. I suggest moving them to Ch. 2. -Forth sentence: Move to evaluation part in Ch. 2. Refine to "The standard deviation of winter-mean DJF SIC shows high variability along the MIZ and the Odden region, in the observational and reanalysis data (Fig. 2)." -Last sentence: Move sentence to evaluation part in Ch. 2. Also, add that the figure shows that TOPAZ4 has a higher interannual variability in the winter-mean DJF sea ice concentration compared with the observations. Add how this discrepancy can or will influence the results and conclusions of this study. Is it exaggerating the interannual variability, thus giving a higher signal to noise-ratio, or is the additional variability not in phase with the observed variability? Please add a time series of the corresponding SIC variability of the observational and reanalysis SIC in the study area as a Fig. 2c. The model was assimilated with SIC data. Mention why it still has discrepancies from the observed SIC data. How often is the model assimilated (daily, monthly?).

Based on the suggestions the manuscript is reorganized accordingly. Kindly note that, TOPAZ4 assimilates data every week. Data assimilation returns an optimal middle solution between the model

and the observations, according to their respective uncertainties, but it should not match exactly the observations. In the assimilation of sea ice concentrations, the sea ice model has a "tighter Marginal Ice Zone (MIZ) than observations" (as stated in the text): a sharper transition zone between the pack ice and the open ocean. After data assimilation, the location of the ice edge is moved closer to observations but the MIZ remains sharper than in observations. Therefore, the sea ice variability in TOPAZ4 is confined to a smaller area than in observations, thus reaching higher percentages in a narrower band. All in all, the TOPAZ4 reanalysis has filtered the observations, in the sense of the Kalman Filter, and gives a higher signal-to-noise ratio than observations, visible with stronger regression values in Figure 3b. The figure will be updated with a colour scale that does not make the variability of TOPAZ4 seem alarmingly high .

Paragraph 140-145: This paragraph is messy, it contains parts of the objective, results, methods and discussion, and it suffers from some poor, vague writing. It needs a strong rewrite, and the sentences that remain in this paragraph needs to be sharp "to the point". -First sentence is about the objective of the paper. Move it to the last paragraph of Ch. 1 and rewrite to a clear sentence. -Sentences 2-4 belong to this paragraph as they present the result and the accompanying brief discussion of the implication of the result, which is appropriate for the chapter. -Fifth and sixth sentences belongs to Ch. 1. I understand that the authors have a need to explain why they move on to investigating the atmospheric fields, but there is too much introductory text here, 'with full sentences simply stating the results of other references. Rather simply add a '. . ., in line with *REFS*' after the fourth sentence. -The last two sentences are a mixture of methods and discussion. I suggest taking them out and, if necessary, starting the following paragraph with a brief mentioning of why the atmospheric influence was investigated. Paragraph 155-175: Very long paragraph, which presents two different figures. Split into two paragraphs where Fig. 5 is presented for the first time. -The first sentence is unnecessary long and difficult to read. Rephrase it. The part "suggests that the large-scale circulation associated with the GSG circulation features a NAO-like meridional pattern although the SLP. . ." could be written as "shows a NAO-like atmospheric pattern associated with the GSG ocean circulation, but with centres of action north of their usual locations (Fig. 4a). "Further, it is not self-explanatory what is meant with the phrase "composite differences of anomalies of ". I understand from reading the paper that you have estimated the difference in winter SLP between the positive and negative Gyre Index periods, using anomalies from the long-term mean, but this should be explicitly written in the methods chapter and also understandable just from reading the figure caption and introduction of the figure in the Ch. 3. If you want to keep the phrase "composite differences of SLP anomalies" then explain its meaning explicitly in the methods. Write which time-period the anomalies are estimated based on, in the text and in the caption. -Sentence starting with "The GSG circulation responds to. . .": I suggest bringing it further up front in the paragraph, as the second sentence. Then say that the correlation coefficient with the static, traditional NAO-pattern is insignificant, showing the importance of taking spatial variability of NAO's centres of action into account. -The two sentences starting with "However, at the same time. . .": It is awkward and needs a rewrite, to e.g. "The associated wind stress can influence sea ice transport through Ekman transport, to- wards the central GS and Greenlands west-coast due Ekman transport" and "We

find anomalous northerly wind stress in the central GS during the positive Gyre Index periods, and vice versa for southerly wind stress." Comment: Is it a southerly wind stress or a weaker northly wind stress during negative Gyre Index periods?

Yes, weakened northly wind stress. All Above suggestions are incorporated in the revised manuscript.

It would be clarifying to show the maps with the mean wind fields for the two different cases, strong and weak gyre. -Panels are introduced in the opposite sequence. Introduce Fig. 5a first, then 5b. -This last part around lines 170-175 contains key discussion that is important and may fit better in a separate paragraph for the broader discussion at the end of Ch. 3. Paragraph 185-190: -The entire paragraph is introductory text and includes what the paper investigates in the third sentence. Move this information to Ch. 1. Paragraph 195-200: -First and second sentences: Rewrite to state the result explicitly, then end with '(Fig. 6a)'. The term 'composite differences' is not very intuitive. Could this be rephrased to something more easily understood? The point is that the figure shows that the ocean temperature in the upper 400 m is higher during the positive Gyre Index periods, right? Please write this simply and explicitly and mention how much higher the temperature is during these periods compared with the mean of the negative Gyre Index periods.

As asked we have added seperate Discussions section.

Third sentence: Remove. This is repetition and another mentioning of introductory text that does not belong to Ch. 3. -Fourth sentence: Rewrite to start with bluntly presenting the result coming from Fig. 6b, i.e. 'There is a significant positive correlation between the Gyre Index and ocean heat transport in the upper 400 m in the smaller study area west of the main study area (r=0.7)'. (The phrase will improve when the two study areas are named and introduced in the same panel in Fig. 1a.) It is cluttering the structure to start with how a previous finding is confirmed here. This is not a proper way to introduce Fig. 6b. Again, stick to the pattern of first presenting the result, and secondly discuss briefly its meaning, implication, mention how it confirms a previous finding or the like. -Last sentence: If this is also for the upper 400 m then it mixes the signatures of variability in the AW with the upper water masses. It would be more appropriate to check separately for different parts of the water column, e.g. 0-100 m and 100-400 m, separately, as AW is typically in the latter whereas the surface waters and polar/ Arctic waters occupies the upper part. That would allow for seeing if the surface water decrease in salinity in periods with negative Gyre Index and more sea ice drift towards the gyre and would give a stronger result on whether AW is influencing higher up in the water column during positive Gyre Index periods. The term 'surfacing' is too strong given that it is likely rising but not necessarily reaching the surface. Be specific about which region this is estimated for.

We thank the reviewer for the above detail constructive suggestions. All suggestions are incorporated in the revised manuscript.

Kindly note that, the salinity is not for the 400m but at the sea surface, the first depth level at the reanalysis data. So the signature of AW to the surface is there. We tried to show in Figure 6b, that with

strong gyre there is increased temperature transport (within AW column~0-400m) in the region indicating warmer AW transport by GSG. The salinity was chosen at the surface level to show that this AW surfaces up in a weakly stratified condition and thus with a strong gyre we get higher salinity at the surface. We have indicated these in the revised manuscript with added analysis to support our claim.

Paragraph 210-215: The paragraph is messy and needs to be tidied up. It starts with discussing figures 4 and 5. Rather start it with 'The Gyre Index is covarying with the Brunt-Väisälä frequency (Fig.7).' And add which depth span of the water column the stratification is estimated over. I suggest continuing with 'Our comparison shows that a weakening of the stratification in the upper part of the water column coincides with a stronger GSG circulation and viceversa.' -Sentence 4: Rewrite to 'This supports that . . . by the GSG can rise under a . . ., hence potentially also the SIC.' -Last sentence: Delete 'further' and 'eastward flowing'. Change 'EGC' to 'JMC'. Delete the clause starting with ', which constitutes' because it is repetition. *A summarizing paragraph with a broader discussion is needed here. This should start with presenting Fig. 8 and write up the complete picture of processes and how the authors interpret their findings.

As suggested we have incorporated all changes and they are explained in a separate discussion section

CH. 4 CONCLUSIONS Please state if atmospheric or oceanic circulation when 'circulation' is mentioned in the conclusion, to avoid confusion. -First sentence: Unclear. Please rewrite. I suggest starting with e.g. 'Here, we investigate . . . and show that . . .'. -Second sentence: add 'the' before 'wind stress curl'. -Third sentence: Rewrite to be more specific, e.g. 'The large-scale atmospheric circulation pattern that influences the GSG circulation resembles a NAO-like pattern with its northern centre of action situated northeast of the typical NAO pattern.' -Fourth sentence: Add a 's' in 'sea ice conditions'. After 'Odden region', add 'in the GS'. Modify end of sentence to 'through Ekman drift of sea ice toward the Greenland coast during periods with northerly winds (Germe et al. 2011).' -Fifth sentence: The sentence is a bit messy. Rewrite it to e.g. 'During periods with anomalously low SLP and strong gyre circulation in the GS, northerly winds and associated Ekman drift causes sea ice drift towards the Greenland coast. This reduces the SIC in the central GS.' -Sixth sentence: Consider a rewrite to 'We show that this is associated with a weakening of the stratification in the upper water column.' or similar, to be more direct and specific. -Seventh sentence: Add a comma after 'into the central GS'. Modify the latter part of the sentence. It has not been shown that the AW reaches all the way to the surface, only that the upper 400 m become warmer and less saline. -Eight sentence: Presentation of Figure 8 is too short and should have been mentioned at the end of Ch. 3 and with a broader discussion of the conceptual view of the findings. -Last sentences after ". . .(Fig. 8)": This part is taking up too much space in the conclusion and gives the impression of dampening the value of the findings of the paper and the closure of the paper becomes too vague. Rather mention this clearly in the methods, that the impact of other smaller scale processes would largely cancel out or act to reduce the correlation coefficients in the processes studied here. But despite these smaller scale processes the results are

significant. However, if mentioned in the conclusion it could be rephrased to e.g. "Despite the presence of smaller scale processes, such as eddies and wave interactions, our results on the larger scale processes are significant with high correlation coefficients. This implies that smaller scale processes largely cancel out over time or are not strong enough to dampen the larger scale processes, at least not when comparing periods with weak and strong gyre circulation in winter when the wind forcing is strong".

The new discussions and conclusion section is written fresh incorporating the suggestions above.

FIGURES
In general, figure captions are lacking information and are not complete, and there are incomplete sentences, e.g. when explaining place names in Fig 1. Make sure all the information is given for each figure. The introduction of study areas should be made up front in Figure 1. In the current version of the manuscript the reader is asked to check for other figures to see which study region is meant. Rather include the two study regions and bathymetry in all the maps were those are needed in order to interpret the results from that figure. Be consistent. Consider to move the larger map from Fig. 4 to Fig. 1, as it fits with zooming in to the study area, and can be referred to in Ch. 1 Introduction. The term "composite differences" is used without further explanation, but is not self explanatory. Please be clear so the reader does not have to check the methods to understand what the figure shows. It can be written out full without too much space, e.g. 'Difference in winter-mean SLP for DJF between time periods with strong and weak Gyre Index during 1991-2017', or something similar.

We take note of this and legends are modified with more self-explanatory details. Also a larger map in Figure 1 is added.

Figure 1: -Consider moving the larger map in Fig. 4a to here to zoom in early on. -Add the other smaller study region as well and name the two. -It is a bit misleading to show the larger study area as a box when in reality is following a bathymetry contour. Rather show it with the 3000 m bathymetry contour. Consider showing it in all the figures that have a map. It should be possible without cluttering the figures. -Add a time series of the Gyre Index as Fig. 1b, and highlight the positive and negative time periods and threshold values 0.75 and −0.75 as e.g. shaded bands and dotted lines. -Highlight the 3000 m isobath contour as e.g. a thicker contour line than the others, to show theregion where the Gyre Index is estimated from.

We have modified Figure 1 addressing the points here.

Figure 2: -This figure belongs to the Methods section, showing the variability of the sea ice concentration fields in winter and is used for evaluation of TOPAZ4. Rather refer to it in Ch. 2. Add label to the colorbar. -Rephrase 'winter (DJF) mean' to 'winter-mean DJF'. -Add more panels to evaluate the model thoroughly. It is important to show that it simulated the ocean stratification and temperature and salinity well.

We have rearranged the figures an dadded more figures for model evaluation.

Figure 3: -Caption: Explanation of the red square is missing. -This can also be part of the evaluation and mentioned for the first time in Ch. 2. Why is the co-variability between SIC and the Gyre Index stronger in TOPAZ4 compared with the observations? How can this influence the results and the interpretation of them? This should be discussed in the methods section. -Fig. 3b: Panel is denoted as (a). -Interpretation of Fig.3: The authors conclude on causality when the figure only shows inverse co-variability. It could be that SIC and the Gyre Index are both affected by the atmospheric wind forcing? Sentence in paragraph 140-145 could be rephrased to e.g.: "This indicate that the GS SIC variability is covarying with the GSG circulation." -Correct the first sentence of the caption to '. . . (a) satellite observations and (b) the TOPAZ4 reanalysis. . .'.

The stronger regression has been explained above: the data assimilation analysis is a Kalman Filter and has less noise compared to observations.

Figure 4: -Show also the mean wind fields of positive and negative Gyre Index periods, not only the anomalies to show the differences between the positive and negative periods. Is the mean of the negative periods a weaker northerly wind field compared to the mean of the positive periods, or is it a southerly wind field, with wind from the south? I assume it is not a mean southerly wind field in the negative periods, but a weaker northerly wind field, and that the anomalies show southerly wind because they are less northerly compared with the temporal mean for the whole study period. But this needs to made clear for the reader. Showing the mean fields would make the interpretations more intuitive and the paper easier to follow. -In the caption, add which time period the anomalies are estimated from. Add which data set the SLP is from. -Consider to add similar maps as b and c for Ekman pumping to make a stronger argument for the "lift of AW" during positive Gyre Index periods. -This figure has different coastlines compared with the other maps. Please be consistent.

The concerened figure is modifed as Figure 6. The lift of AW or higher temperature in upeer ocean due to vertical mixing is shown in Figure 9.

Figure 5: -In caption, delete 'vectors'. -Again, showing only the anomalies means it is not entirely clear if the positive Gyre Index periods are associated with weaker southward sea ice drift or if the mean sea ice drift is from the other direction, i.e., northward sea ice drift. It would be more intuitive and easier to follow the manuscript if the mean field is shown for sea ice drift in positive and negative Gyre Index periods.

Kindly note that, Figure 5a is the climatological ice vector fields, while 5b is regression of ice vectors on Gyre index. They are now Figure 7a,b.

Figure 6: -Panel 6a: Add outlines for the study areas from which the variables in 6b are estimated from. -Panel 6b: Add line for y=0. Add bands of shading showing the positive and negative Gyre Index periods, from which the map in 6a is estimated from. Consider adding a similar map as that of panel 6a for the upper 100 m or even the upper 50 m. This could be very interesting and give information that helps make the interpretation of the complete picture of processes (i.e., regards Ekman transport of surface water). -First sentence in caption is unclear. Rephrase to 'Difference in average potential temperature anomalies in the upper 400 m of the water column between positive and negative Gyre Index periods during 1991–2017.' -The term 'temperature advection' is perhaps better phrased as 'heat transport'. Explain in the methods how it was estimated. -Second sentence in caption is unclear. Rephrase to 'Time series of the Gyre Index (blue curve) and standardized anomalies of the salinity and tempera- ture advection in the upper 400 m.' Please do not use the term 'surface salinity' for the salinity in the upper 400 m, as 'surface' is typically associated with the upper 0-50 m or so.

We have modified the analysis, figures to clearly present as it is suggested. Kindly note that, In Figure 6b, we showed salinity at the surface level only not in upper 400m. We tried to show in Figure 6b, that with strong gyre there is increased temperature transport (within AW column~0-400m) in the region indicating warmer AW transport by GSG. The salinity was chosen at the surface level to show that this AW vertically mixes up up in a weakly stratified condition and thus with a strong gyre we get higher salinity at the surface.

Figure 7: -Add similar panels for temperate and salinity in ocean and increase the depth span to show the lift of AW. This can help justify the conclusions regards thelifting of AW in periods with positive Gyre Index. -Why is the Brunt-Väisälä frequency not estimated for the same region as the Gyre Index? Using the other smaller region outside the gyre makes the interpretation harder. The main response regards the lifting of the interface between the upper polar water masses and AW is occurring most strongly in the centre of the gyre?

We have added the analysis with temperature and salinity. The main reason for selecting a different region is shown in Figure 2 and 5. While the gyre circulation changes in the central GS, the effect of it on sea ice is most realized in the MIZ, where the Brunt-Väisälä frequency is shown. We intend to focus the on gyre's impact on the SIC. In the centre of gyre we dont get any clear signal that says the gyre could affect the sea ice there. Also the region in MIZ shows maximum interannual variability in both observation and model. Note that, allthough they differ in magnitude but for understanding the processes the similar pattern of significant influence is compelling.

Figure 8: -Flip the diagram on the side, with the atmospheric pathway on top and the oceanic pathway below. -Add schematics of how the authors interpret the process in the horizontal (showing divergence of sea ice and freshwater due to Ekman transport in response to stronger wind forcing and related

increased AW recirculation and inflow to the GS) and vertically (showing the Ekman pumping and lift of the AW in response).

Based on the suggestion the new figure is preapred and shown as Figure 10 in the revised manuscript.

USAGE OF TERMS -use the term "winter-mean (DJF)" instead of "winter time (DJF)". Avoid abbreviations as much as possible for easy reading. -the term "northerly" is used for both wind direction and sea ice drift. To avoid confusion, consider using "northerly" and "southerly" only for wind, and "northward" for sea ice drift and oceanic currents. See e.g. second sentence at the beginning of page 8, where the usage of "northerly" for both wind and sea ice drift is confusing. It is not clear if the sea ice drift is from the north or from the south from this sentence. -use apostrophes only when introducing a new term, like 'the Gyre Index'. Then refer to it simply as the Gyre Index without apostrophes on later mentions. The same goes for the Odden region. -Present the two study regions in the methods section, new Ch. 2.2., in Fig. 1a and Fig. 1 captions. Name the two study regions and use these names consistently throughout the paper. Write in the methods what you have estimated for each study area. -Which depth range is the surface salinity anomaly estimated for? Write this in the methods. If this is the upper 400 m then "upper ocean salinity" is more appropriate. -The term 'validation' is used in section 2.2. 'Evaluation' is a better suited term because it reflects that all models have strengths and weaknesses, no models are perfect, and a key point is to make sure that the model results are useful for investigating the objective of the paper with the chosen approach.

We thank the reviewer for his in details evaluation of the manuscript. We strongly believe this suggestions will improve our manuscript's presentation and readbility to a great extent. We have incorporated the suggestions made and/or modify the figures, analysis, presentation style so that the points raised here are adequetly addressed.

---

## Referee Report (RR1)

**The impact of atmospheric and oceanic circulations on the Greenland Sea ice concentration**

by Sourav Chatterjee, Roshin P. Raj, Laurent Bertino, Sebastian H. Mernild, Nuncio Murukesh, and Muthalagu Ravichandran

The revised version of the manuscript is a substantial improvement on the original submission. For the most part I think that my concerns have been well addressed, but I still have some comments regarding the TOPAZ-EN4 comparison and the upper-ocean stratification in the western Greenland Sea that I hope the authors will take into consideration.

**Specific comments:**

Hovmöller plots of SST and SSS and time series of potential temperature, salinity, and stratification from the western Greenland Sea region are shown in Figs. 3 and 4. The data are taken from the TOPAZ reanalysis and from the EN4 objective analysis from the period 1991 to 2017. The authors note that the temporal evolution of these parameters appears to be well represented in TOPAZ, but that there are some differences such as the magnitude of the stratification and the location of the polar front separating the cold, fresh polar waters along Greenland from the warmer, saltier Arctic waters in the Greenland Sea. My concern is that there are hardly observations from the western Greenland Sea region chosen for this comparison (consider, for example, Fig. 2a in Brakstad *et al.*, 2019). It appears that EN4 will relax to climatology when there are no observations (Good *et al.*, 2013). The western Greenland Sea is an exceptionally data-sparse region, so I am very sceptical that these two figures provide much information about how well TOPAZ represents the hydrographic conditions in this region. I suggested instead that TOPAZ is evaluated against observations in the central Greenland Sea and that Latarius and Quadfasel (2016) or Brakstad *et al.* (2019) would be good points of comparison. In my opinion that would be a more sensible evaluation of TOPAZ.

I have some concerns about the upper ocean buoyancy frequency (Fig. 9a) in the western Greenland Sea. The western Greenland Sea region that you have defined is a region of complex hydrography. It includes the Greenland shelf, which is dominated by polar surface water, the shelf break and upper slope where the EGC transports Atlantic-origin water at intermediate depth, and the interior Greenland Sea where Arctic-origin water is formed (e.g. Håvik *et al.*, 2017; Renfrew *et al.*, 2019). I don't think that a spatial average across this region is very meaningful. The Greenland shelf is highly stratified, while offshore of the polar front the stratification is far weaker and deep convection is possible. In my opinion stratification averaged across this region is not a robust measure of vertical mixing of Atlantic-origin water or of the strength of the Greenland Sea Gyre.

Line 26:

The importance of the Greenland Sea to the AMOC is still overstated. While recent results of Huang *et al.* (2020) indicate that the Greenland Sea is an important source of the densest overflow waters from the Nordic Seas in the present climate and Chafik and Rossby (2019) demonstrate that the overflows from the Nordic Seas are key to the lower limb of the Atlantic Meridional Overturning Circulation, none of the references cited in the paper (or any other papers that I am aware of) demonstrate that the strength of the overturning circulation partly depends on the amount of freshwater in the Greenland Sea.

Line 42:

This sentence is unclear. Ice free conditions, or at least partially ice free conditions, are required for deep convection to occur. According to Moore *et al.* (2015) the reduced depth of convection in the central Greenland Sea is not because it is ice free - that is a prerequisite for deep convection. It is instead caused by the retreat of the ice edge, and the region of strongest ocean-to-atmosphere fluxes which is tied to the ice edge, toward Greenland such that heat loss from the central Greenland Sea is reduced.

Lines 53-55:

I counted 6 different acronyms on these three lines. Are all of these really necessary? The readability of the text would improve if the number of acronyms were reduced.

Line 57:

The subpolar North Atlantic is generally considered the subpolar gyre south of the Greenland-Scotland Ridge, not the Nordic Seas.

Line 147:

The region outlined in Fig. 2 is in the very southwestern part of the Greenland Sea (the Iceland Sea is more or less immediately to the south of this region). As such, I think referring to it as southwestern Greenland Sea rather than western Greenland Sea would be more appropriate.

Line 219:

The term "temperature advection" should be defined also in the text, not only in the caption of Fig. 8.

Line 246:

How often is this pattern, which resembles the NAO but has centers of action shifted toward the north, realized?

Figure 1:

The thick black line marking the 3000 m isobath is not clearly visible.

Figure 3:

It would be good to specify in the caption that monthly values are considered.

Figure 10:

Ekman is misspelled in box A2.

**References**

Brakstad A, Våge K, Håvik L, Moore GWK. 2019. Water mass transformation in the Greenland Sea during the period 1986-2016. *Journal of Physical Oceanography* **49**: 121–140, doi:10.1175/JPO–D–17–0273.1.

Chafik L, Rossby T. 2019. Volume, heat, and freshwater divergences in the Subpolar North Atlantic suggest the Nordic Seas as key to the state of the Meridional Overturning Circulation. *Geophysical Research Letters* **46**: doi:10.1029/2019GL082 110.

Good SA, Martin MJ, Rayner NA. 2013. EN4: Quality controlled ocean temperature and salinity profiles and monthly objective analyses with uncertainty estimates. *Journal of Geophysical Research: Oceans* **118**: doi:10.1002/2013JC009 067.

Håvik L, Pickart RS, Våge K, Thurnherr AM, Beszczynska-Möller A, Walczowski W, von Appen WJ. 2017. Evolution of the East Greenland Current from Fram Strait to Denmark Strait: Synoptic measurements from summer 2012. *Journal of Geophysical Research: Oceans* : doi:10.1002/2016JC012 228.

Huang J, Pickart RS, Huang RX, Lin P, Brakstad A, Xu F. 2020. Sources and upstream pathways of the densest overflow in the Nordic Seas. *Nature Communications* : submitted for publication.

Latarius K, Quadfasel D. 2016. Water mass transformation in the deep basins of the Nordic Seas: Analyses of heat and freshwater budgets. *Deep Sea Research I* **114**: 23–42, doi:10.1016/j.dsr.2016.04.012.

Moore GWK, Våge K, Pickart RS, Renfrew IA. 2015. Open-ocean convection becoming less intense in the Greenland and Iceland Seas. *Nature Climate Change* **5**: doi:10.1038/nclimate2688.

Renfrew IA, Pickart RS, Våge K, Moore GWK, Bracegirdle T, Elvidge AD, Jeansson E, Lachlan-Cope T, Papritz L, Reuder J, Sodemann H, Terpstra A, Waterman S, Valdimarsson H, Weiss A, Almansi M, Bahr F, Brakstad A, Barrell C, Brooke JK, Brooks BJ, Brooks IM, Brooks ME, Bruvik EM, Duscha C, Fer I, Golid HM, Hallerstig M, Hessevik I, Huang J, Houghton L, Jónsson S, Jonassen M, Jackson K, Kvalsund K, Kolstad EW, Konstali K, Kristiansen J, Ladkin R, Lin P, Macrander A, Mitchell A, Olafsson H, Pacini A, Payne C, Palmason B, Pérez-Hernández MD, Peterson AK, Petersen GN, Pisareva MN, Pope JO, Seidl A, Semper S, Sergeev D, Skjelsvik S, Søiland H, Smith D, Spall MA, Spengler T, Touzeau A, Tupper G, Weng Y, Williams K, Yang X, Zhou S. 2019. The Iceland Greenland Seas Project. *Bulletin of the American Meteorological Society* **100**: 1795–1817, doi:10.1175/BAMS–D–18–0217.1.

---

## Author Response (AR2)

The impact of atmospheric and oceanic circulations on the Greenland Sea ice concentration
by Sourav Chatterjee, Roshin P. Raj, Laurent Bertino, Sebastian H. Mernild, Nuncio Murukesh, and Muthalagu Ravichandran

The revised version of the manuscript is a substantial improvement on the original submission. For the most part I think that my concerns have been well addressed, but I still have some comments regarding the TOPAZ- EN4 comparison and the upper-ocean stratification in the western Greenland Sea that I hope the authors will take into consideration.

Authors' reply:

The authors thank the reviewer for appreciating the modifications made in the revised version. The concerns mentioned here are addressed below:

Specific comments:

Hovmöller plots of SST and SSS and time series of potential temperature, salinity, and stratification from the western Greenland Sea region are shown in Figs. 3 and 4. The data are taken from the TOPAZ reanalysis and from the EN4 objective analysis from the period 1991 to 2017. The authors note that the temporal evolution of these parameters appears to be well represented in TOPAZ, but that there are some differences such as the magnitude of the stratification and the location of the polar front separating the cold, fresh polar waters along Greenland from the warmer, saltier Arctic waters in the Greenland Sea. My concern is that there are hardly observations from the western Greenland Sea region chosen for this comparison (consider, for example, Fig. 2a in Brakstad et al., 2019). It appears that EN4 will relax to climatology when there are no observations (Good et al., 2013). The western Greenland Sea is an exceptionally data-sparse region, so I am very sceptical that these two figures provide much information about how well TOPAZ represents the hydrographic conditions in this region. I suggested instead that TOPAZ is evaluated against observations in the central Greenland Sea and that Latarius and Quadfasel (2016) or Brakstad et al. (2019) would be good points of comparison. In my opinion that would be a more sensible evaluation of TOPAZ.

We have followed the suggestions provided by the reviewer. We found TOPAZ4 doesn't perform well in the central GS in terms long term trend, particularly in the deeper levels. This is probably due to the fact that TOPAZ has been initialized with too warm in deep waters of this area (Xie et al., 2017). Please find the comparison below (Figure 1), where EN4 reproduces the deep warming as found in Latarius & Quadfasel (2016) observations. But the TOPAZ4 reanalysis seems to recover from a too warm initial condition at depths of 400m-1600m. The warm bias is however not visible in the upper 100 meters of the water column, which is where our analysis is focusing.

In addition, we note that the processes at play are taking place further in the south-western Greenland Sea, where the model quality appears to be fair in comparison with climatology and also with other reanalysis (ORAS5). Note that in other areas in the Nordic Seas, fore.gFram Strait or Svinøy sections also TOPAZ4 performs reasonably well (Chatterjee et al., 2018), although with local biases. Further the oceanic processes explained here are in the south-western Greenland Sea and its impact on sea ice variability therein is consistent with the satellite sea ice observations. Since our major focus area is the south-western Greenland Sea, we think that it is more meaningful to compare TOPAZ4 with the limited existing data sources there (where the different processes are explained and their impact on sea ice is found to be consistent), than comparing in a different

region, particularly when it is known that the performance of TOPAZ4 may vary regionally (Sakov et al., 2012, Xie et al., 2017).

[Figure]

*Figure 1: Potential Temperature in Greenland Sea gyre (top) from Latarius and Quadfasel (2016) (middle) EN4 (c) TOPAZ4.*

We thank the reviewer for raising the issue and we do feel that it needs to be addressed in the main text. Accordingly, we have added the following lines in the discussion (line 296-301 ):

*It should be noted that the complex subsurface processes and their interactions with the large scale circulation are often difficult to capture in the reanalysis, particularly with sporadic subsurface observations in both time and space. For example, while the surface variables are well captured in TOPAZ4, the reanalysis is too warm in the GS below 300 m as observed in Xie et. al, 2017 (their Figure 9). Of particular interest in this study, the south-western GS, is a particularly sparse region in observational data. Increased long-term observations from these areas would help improving the reanalysis datasets and better understand the complex atmosphere-ocean interactions and their impact on the sea ice variability of this region.*

I have some concerns about the upper ocean buoyancy frequency (Fig. 9a) in the western Greenland Sea. The western Greenland Sea region that you have defined is a region of complex hydrography. It includes the Greenland shelf, which is dominated by polar surface water, the shelf break and upper slope where the EGC transports Atlantic-origin water at intermediate depth, and the interior Greenland Sea where Arctic-origin water is formed (e.g. Håvik et al., 2017; Renfrew et al., 2019). I don't think that a spatial average across this region is very meaningful. The Greenland shelf is highly stratified, while offshore of the polar front the stratification is far weaker and deep convection is possible. In my opinion stratification averaged across this region is not a robust measure of vertical mixing of Atlantic-origin water or of the strength of the Greenland Sea Gyre.

We agree that the hydrography changes in this region within small horizontal extent and that makes the spatial average over the region bit ambiguous. Thank you for raising this issue! We checked it further and the below analysis shows that it does not affect the main message of the study.

The main purpose of showing the stratification is to highlight that indeed there is weakening of stratification in this region and that causes the Atlantic waters to mix vertically, which is further supported by Fig 9b and c.

It is this process, which we are attempting to attribute to the sea ice variability of this whole region and associate with the Gyre circulation. One would expect, and also as evident in the figure below, the extent/magnitude of the impact of Gyre circulation on the sea ice concentration (through stratification changes) depends on the complex hydrography of the region as rightly mentioned by the reviewer. For example, it can be noticed that during strong/weak gyre circulation years (negative/positive values), the spatial pattern of the satellite observed sea ice concentrations closely follows the stratification pattern. The impact on sea ice is larger in the eastern side with less stratified waters and it is smaller in the highly stratified Greenland coast waters as can be expected. Nonetheless, the main point is that the process is valid for both types of hydrography, only the magnitude of the implication of the process differs.

We have kept the text identical for the sake of brevity.

[Figure]

Figure 2: Logarithm of squared Brunt-Väisälä Frequency (left) , Sea ice concentration (middle) and Gyre Index (right) during the DJF.

Line 26:
The importance of the Greenland Sea to the AMOC is still overstated. While recent results of Huang et al.(2020) indicate that the Greenland Sea is an important source of the densest overflow waters from the Nordic Seas in the present climate and Chafik and Rossby (2019) demonstrate that the overflows from the Nordic Seas are key to the lower limb of the Atlantic Meridional Overturning Circulation, none of the references cited in the paper (or any other papers that I am aware of) demonstrate that the strength of the overturning circulation partly depends on the amount of freshwater in the Greenland Sea.

We have modified the text following the suggestion as below (Line 26-27):

The freshwaters in the GS plays an important part for Nordic Seas overflow (Huang et al., 2020), which constitutes the lower limb of the Atlantic meridional overturning circulation (Chafik and Rossby 2019).

Line 42:
This sentence is unclear. Ice free conditions, or at least partially ice free conditions, are required for deep convection to occur. According to Moore et al. (2015) the reduced depth of convection in the central Greenland Sea is not because it is ice free - that is a prerequisite for deep convection. It is

instead caused by the retreat of the ice edge, and the region of strongest ocean-to-atmosphere fluxes which is tied to the ice edge, toward Greenland such that heat loss from the central Greenland Sea is reduced.

*We agree that the sentence is confusing. The sentence is removed in the revised version.*

Lines 53-55:

I counted 6 different acronyms on these three lines. Are all of these really necessary? The readability of the text would improve if the number of acronyms were reduced.

*We have reduced few acronyms as below (Line # 52-55) :*

*Selyuzhenok et al. (2020) also argued that consistent positive North Atlantic Oscillation (NAO) forcing in recent decades have led to warmer AW in the Nordic Seas and resulted in a declining sea ice volume trend. However, the response of Nordic Seas circulation to the atmospheric forcing and the mechanism through which it can influence the SIC in GS is not studied in detail.*

Line 57:

The subpolar North Atlantic is generally considered the subpolar gyre south of the Greenland-Scotland Ridge, not the Nordic Seas.

*Modified as: (Line 57-58)*

*The Greenland Sea Gyre (GSG) is a prominent large-scale feature of the Nordic Seas circulation and can be identified as a cyclonic circulation in the central GS basin (Fig. 1).*

Line 147:

The region outlined in Fig. 2 is in the very southwestern part of the Greenland Sea (the Iceland Sea is more or less immediately to the south of this region). As such, I think referring to it as southwestern Greenland Sea rather than western Greenland Sea would be more appropriate.

*In the revised manuscript the region is addressed as southwestern Greenland Sea.*

Line 219:
The term "temperature advection" should be defined also in the text, not only in the caption of Fig. 8.

*Defined (line # 218)*

Line 246:
How often is this pattern, which resembles the NAO but has centers of action shifted toward the north, realized?

*The shift of NAO centre of action has been identified in many studies earlier (e.g Zhang et al., 2008, Moore et al., 2013). The shift is more prominent from late 1990s as observed in the figure from Zhang et al., 2008.*

[Figure]

*Figure 3: (a) Positions of the centers of action and (b) the first EOF/PC spatial patterns in the recent representative time windows. The circles and triangles represent positions of the centers of action for each time window centered in the year shown in the color bar over the Arctic, North Atlantic, and North Pacific. From Zhang et al., 2008:*

Figure 1:
The thick black line marking the 3000 m isobath is not clearly visible.

Modified

Figure 3:
It would be good to specify in the caption that monthly values are considered.

Mentioned

Figure 10:
Ekman is misspelled in box A2.

Corrected

---

## Author Response (AR3)

tc-2020-127

The impact of atmospheric and oceanic circulations on the Greenland Sea ice concentration
by Sourav Chatterjee, Roshin P. Raj, Laurent Bertino, Sebastian H. Mernild, Nuncio Murukesh, and Muthalagu Ravichandran

The revised version of the manuscript is a substantial improvement on the original submission. For the most part I think that my concerns have been well addressed, but I still have some comments regarding the TOPAZ- EN4 comparison and the upper-ocean stratification in the western Greenland Sea that I hope the authors will take into consideration.

Authors' reply:

The authors thank the reviewer for appreciating the modifactions made in the revised version. The concerns mentioned here are addressed below:

Specific comments:

Hovmöller plots of SST and SSS and time series of potential temperature, salinity, and stratification from the western Greenland Sea region are shown in Figs. 3 and 4. The data are taken from the TOPAZ reanalysis and from the EN4 objective analysis from the period 1991 to 2017. The authors note that the temporal evolution of these parameters appears to be well represented in TOPAZ, but that there are some differences such as the magnitude of the stratification and the location of the polar front separating the cold, fresh polar waters along Greenland from the warmer, saltier Arctic waters in the Greenland Sea. My concern is that there are hardly observations from the western Greenland Sea region chosen for this comparison (consider, for example, Fig. 2a in Brakstad et al., 2019). It appears that EN4 will relax to climatology when there are no observations (Good et al., 2013). The western Greenland Sea is an exceptionally data-sparse region, so I am very sceptical that these two figures provide much information about how well TOPAZ represents the hydrographic conditions in this region. I suggested instead that TOPAZ is evaluated against observations in the central Greenland Sea and that Latarius and Quadfasel (2016) or Brakstad et al. (2019) would be good points of comparison. In my opinion that would be a more sensible evaluation of TOPAZ.

We found TOPAZ4 doesn't perform well in the central GS in terms long term trend, particularly in the deeper levels. This is probably due to the fact that TOPAZ has too deep mixing in this area (Xie et al., 2017). Please find the below comparison.

However, please note that the proceses we are talking about is in the south-western Greenland Sea, where it appears to be fair in comparison with available observation and also with other reanalysis (ORAS5). Note that even in other areas e.g Fram Strait or Svinoy sections also TOPAZ4 performs reasonably well (Chatterjee et al., 2018), with somewhat systematic bias. Further the oceanic processes explained here in the south-western Greenland Sea and its impact on sea ice variability therein is consistent with the satellite sea ice observations. Since our major focus area is south-western Greenland Sea, we think that it is more meaningful to compare TOPAZ4 with limited existing data sources in this region (where the different processes are explained and their impact on sea ice is found to be consistent), than comparing it in a different region, particularly when it is known that the performance of TOPAZ4 may vary at locations (Sakov et al., 2012, Xie et al., 2017).

[Figure]

Figure 1: Potential Temperature in Grrenland Sea gyre (top) from Latarius and Quadfasel (2016) (middle) EN4 (c) TOPAZ4.

We thank the reviewer for raising the issue and we do feel that it needs to be addressed in the main text. Accordingly, we have added the following lines in the discussion:

*However it should be noted that the complex subsrface processes and their interactions with large scale circulation are often difficult to capture in the reanalysis, particularly with sparse and interrupted subsurface observations over time and space. For example, while the surface variables are well captured in TOPAZ4, it has some limitations with the subsurface properties as observed in Xie et. al, 2017. Of particular interest in this study, the south-western GS, is an exceptionally observational data sparse region. Increased long-term observations from these areas will be helpful in improvement of the reanalysis datasets and better understanding of the complex atmosphere-ocean interaction processes and their impact on the sea ice variability of this region.*

I have some concerns about the upper ocean buoyancy frequency (Fig. 9a) in the western Greenland Sea. The western Greenland Sea region that you have defined is a region of complex hydrography. It includes the Greenland shelf, which is dominated by polar surface water, the shelf break and upper slope where the EGC transports Atlantic-origin water at intermediate depth, and the interior Greenland Sea where Arctic-origin water is formed (e.g. Håvik et al., 2017; Renfrew et al., 2019). I don't think that a spatial average across this region is very meaningful. The Greenland shelf is highly stratified, while offshore of the polar front the stratification is far weaker and deep convection is possible. In my opinion stratification averaged across this region is not a robust measure of vertical mixing of Atlantic-origin water or of the strength of the Greenland Sea Gyre.

We agree that the hydrography changes in this region within small horizontal extent and that makes the spatial average over the region bit ambiguous. Thank you for raising this issue! We checked it further and the below analysis shows that it does not affect the main message of the study.

The main purpose of showing the startification is to highlight that indeed there is weakening of stratification in this region and that causes the Atlantic waters to mix vertically , which is further supported by Fig 9b and c.

It is this process which we are attempting to attribute to the sea ice variability of this whole region and associate with the Gyre circulation. One would expect and also as evident in the figure below, the extent/magnitude of the impact of Gyre circulation on the sea ice concentration (through startification changes) depends on the complex hydrography of the region as rightly mentioned by the reviewer. For example, it can be noticed that during strong/weak gyre circulation years (negative/positive values), the spatial pattern of the satellite observed sea ice concetration closely follows the startification pattern. The impact on sea ice is more in the eastern side with less startified waters and it is less in the highly startified Greenland coast waters as can be expected. Nonetheless, the main point is that the process is valid for both types of hydrography, only the magnitude of the implication of the process differs.

[Figure]

Line 26:
The importance of the Greenland Sea to the AMOC is still overstated. While recent results of Huang et al.(2020) indicate that the Greenland Sea is an important source of the densest overflow waters from the Nordic Seas in the present climate and Chafik and Rossby (2019) demonstrate that the overflows from the Nordic Seas are key to the lower limb of the Atlantic Meridional Overturning Circulation, none of the references cited in the paper (or any other papers that I am aware of) demonstrate that the strength of the overturning circulation partly depends on the amount of freshwater in the Greenland Sea.

We have modified the text following the suggestion as below:

*The freshwaters in the GS plays an important part for Nordic Seas overflow (Huang et al., 2020), which constitutes the lower limb of the Atlantic meridional overturning circulation (Chafik and Rossby 2019).*

Line 42:
This sentence is unclear. Ice free conditions, or at least partially ice free conditions, are required for deep convection to occur. According to Moore et al. (2015) the reduced depth of convection in the central Greenland Sea is not because it is ice free - that is a prerequisite for deep convection. It is instead caused by the retreat of the ice edge, and the region of strongest ocean-to-atmosphere fluxes

which is tied to the ice edge, toward Greenland such that heat loss from the central Greenland Sea is reduced.

The sentence is removed in the revised version.

Lines 53-55:

I counted 6 different acronyms on these three lines. Are all of these really necessary? The readability of the text would improve if the number of acronyms were reduced.

We have reduced few  acronyms as below:

*Selyuzhenok et al. (2020) also argued that consistent positive North Atlantic Oscillation (NAO) forcing in recent decades have led to warmer AW in the Nordic Seas and resulted in a declining sea ice volume trend. However, the response of Nordic Seas circulation to the atmospheric forcing and the mechanism through which it can influence the SIC in GS is not studied in detail.*

Line 57:

The subpolar North Atlantic is generally considered the subpolar gyre south of the Greenland-Scotland Ridge, not the Nordic Seas.

Modified as:

*The Greenland Sea Gyre (GSG) is a prominent large-scale feature of the Nordic Seas circulation and can be identified as a cyclonic circulation in the central GS basin (Fig. 1).*

Line 147:

The region outlined in Fig. 2 is in the very southwestern part of the Greenland Sea (the Iceland Sea is more or less immediately to the south of this region). As such, I think referring to it as southwestern Greenland Sea rather than western Greenland Sea would be more appropriate.

In the revised manuscript the region is addressed as  southwestern Greenland Sea

Line 219:
The term "temperature advection" should be defined also in the text, not only in the caption of Fig. 8.

Defined

Line 246:
How often is this pattern, which resembles the NAO but has centers of action shifted toward the north, realized?

The shift of NAO centre of action has been identified in many stuides earlier (e.g Zhang et al., 2008, Moore et al., 2013). The shift is more prominet from late 19902 as observed in the figure from Zhang et al., 2008.

[Figure]

Figure 3: (a) Positions of the centers of action and (b) the first EOF/PC spatial patterns in the recent representative time windows. The circles and triangles represent positions of the centers of action for each time window centered in the year shown in the color bar over the Arctic, North Atlantic, and North Pacific. From Zhang et al., 2008:

Figure 1:
The thick black line marking the 3000 m isobath is not clearly visible.

Modified

Figure 3:
It would be good to specify in the caption that monthly values are considered.

Mentioned

Figure 10:
Ekman is misspelled in box A2.

Corrected